# Feline infectious peritonitis epizootic caused by a recombinant coronavirus

Charalampos Attipa[1,2,3,4,9 ✉], Amanda S. Warr[2,9 ✉], Demetris Epaminondas[5], Marie O'Shea[2], Andrew J. Hanton[2], Sarah Fletcher[2], Alexandra Malbon[1], Maria Lyraki[6], Rachael Hammond[1], Alexandros Hardas[7], Antria Zanti[3], Stavroula Loukaidou[3], Michaela Gentil[8], Danielle Gunn-Moore[1], Samantha J. Lycett[2], Stella Mazeri[2] & Christine Tait-Burkard[2 ✉]

Cross-species transmission of coronaviruses (CoVs) poses a serious threat to both animal and human health[1–3]. While the large RNA genome of CoVs shows relatively low mutation rates, recombination within genera is frequently observed[4–7]. Companion animals are often overlooked in the transmission cycle of viral diseases; however, the close relationship of feline (FCoV) and canine CoV (CCoV) to human hCoV-229E[5,8], as well as the susceptibility of these animals to SARS-CoV-2[9], highlight their importance in potential transmission cycles. While recombination between CCoV and FCoV of a large fragment spanning *orf1b* to *M* has been previously described[5,10], here we report the emergence of a highly pathogenic FCoV–CCoV recombinant responsible for a rapidly spreading outbreak of feline infectious peritonitis (FIP) originating in Cyprus[11]. The minor recombinant region, spanning spike (*S*), shows 96.5% sequence identity to the pantropic canine coronavirus NA/09. Infection has rapidly spread, infecting cats of all ages. Development of FIP appears to be very frequent and sequence identities of samples from cats in different districts of the island are strongly supportive of direct transmission. A near-cat-specific deletion in the domain 0 of *S* is present in more than 90% of cats with FIP. It is unclear as yet whether this deletion is directly associated with disease development, and it may be linked to a biotype switch[12]. The domain 0 deletion and several amino acid changes in S, particularly the receptor-binding domain, indicate potential changes to receptor binding and cell tropism.

After two epidemics, SARS-CoV (2002–2004) and MERS-CoV (2012–ongoing), and a pandemic of previously unseen proportions, SARS-CoV-2 (2019–onwards), CoVs no longer need lengthy introductions of importance and scale. They are not only present in the human population but also in wildlife[13–15], companion animals[16–19] and livestock[13,20,21], and these viruses have major impacts in all species. The innate ability of CoVs to recombine with other CoVs continues to fuel their species-switching ability. It is therefore not surprising that both human and animal CoVs are linked in complex transmission and evolution cycles[3,14,19,22].

FCoV is found across the globe. The virus exists in two biotypes with the main biotype, feline enteric coronavirus (FECV), showing low virulence and clinical signs that are atypically limited to mild enteritis. The second biotype of FCoV, which was proposed to originate each time from a mutation in an FECV-infected cat (reviewed previously[23]), is known as feline infectious peritonitis virus (FIPV). FIPV causes FIP, which is a fatal disease if left untreated. Clinical signs include abdominal swelling due to peritoneal fluid, fever, weight loss, lethargy, anorexia, dyspnoea, ocular abnormalities and neurological signs[8,16,24,25]. Mutations in *S* or the accessory genes *3abc* and *7ab* of FCoV[8,16,23,26] are thought to result in changes to the virus's tropism from cells in the enteric tract

to macrophages, resulting in the different disease presentation seen with the two biotypes. This change in primary tropism also affects the virus's ability to transmit from cat to cat, with the main transmission pathway of FECV being faecal–oral and FIPV typically having relatively poor transmission potential. Antivirals, including remdesivir and GS-441524, have recently been successfully used to treat cats with FIP[27].

FCoV and CCoV both belong to the *Alphacoronavirus suis* (previously known as *Alphacoronavirus 1*) species alongside the porcine transmissible gastroenteritis virus (TGEV) and, probably, the rabbit enteric coronavirus, which has never been fully sequenced[28,29]. Both FCoV and CCoV have evolved two different serotypes through complex recombination events between the two viruses with a suspected stepwise evolution from CCoV-2 to TGEV and the later *S* deletion to porcine respiratory coronavirus (PRCV)[23,30]. While recombination events between CCoV and FCoV have substantially contributed to the serotype evolution and have been frequently described, to date, none of them led to enhanced transmissibility of FIP beyond closest contact[10,23,31,32]. Similarly, recombination events have been reported between CCoV and TGEV[33], the latter of which has been found to recombine with the *Pedacovirus* (*Alphacoronavirus* genus) porcine epidemic diarrhoea

[1]Royal (Dick) School of Veterinary Studies, University of Edinburgh, Easter Bush, Midlothian, UK. [2]The Roslin Institute, Royal (Dick) School of Veterinary Studies, University of Edinburgh, Easter Bush, Midlothian, UK. [3]Vet Dia Gnosis, Limassol, Cyprus. [4]Centre for Inflammation Research, Institute for Regeneration and Repair, The University of Edinburgh, Edinburgh, UK. [5]Veterinary Services, Ministry of Agriculture, Natural Resources and Environment, Nicosia, Cyprus. [6]Plakentia Veterinary Clinic, Athens, Greece. [7]Department of Pathobiology and Population Sciences, Royal Veterinary College, Hatfield, UK. [8]Laboklin, Bad Kissingen, Germany. [9]These authors contributed equally: Charalampos Attipa, Amanda S. Warr. ✉e-mail: charalampos.attipa@ed.ac.uk; amanda.warr@roslin.ed.ac.uk; christine.burkard@roslin.ed.ac.uk

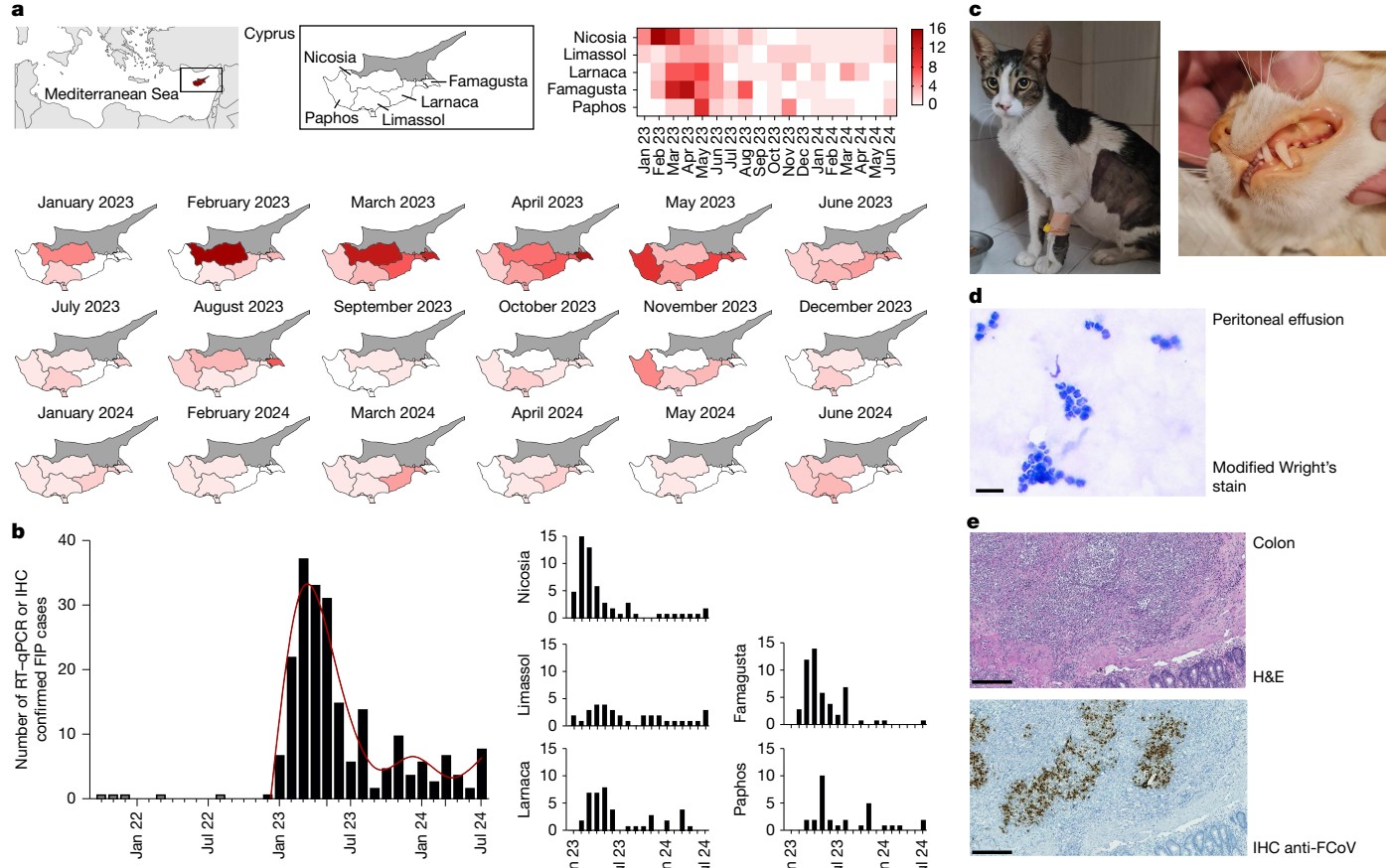

**Fig. 1 | Epidemiology and pathology of the FIP outbreak in Cyprus, January 2023 to June 2024. a**, The distribution of RT–qPCR or IHC-confirmed FIP cases across Cyprus. The first image shows Cyprus within the Eastern Mediterranean. The darker colours indicate higher numbers of confirmed cases over time within each district as highlighted in the overview heat map with key. The maps were generated using mapchart.net. $n = 215$. **b**, RT–qPCR/IHC-confirmed case rates resolved by time and province. A 6 knot spline interpolation highlights the three waves observed to date. **c**, Clinical presentation of cats with FIP due to FCoV-23. Left, a cat with the effusive form of FIP showing abdominal distention due to peritoneal effusion, an unkept coat, low body-condition score and poor muscle condition. Right, a cat presenting with jaundice evidenced by yellow/orange discoloration of the mucous membranes and mucocutaneous junctions. Images courtesy of E. Georgiadi. **d**, Representative peritoneal effusion smear photomicrograph from one Cypriot cat with confirmed FIP due to FCoV-23 infection. Non-degenerative neutrophils are present on a protein-rich background shown using a modified Wright's stain. Scale bar, 20 µm. **e**, Representative photomicrographs showing a section of colonic mucosa and submucosa from one cat with confirmed FIP due to FCoV-23 infection. Top, haematoxylin and eosin (H&E)-stained histology section showing coalescing infiltration of predominantly the submucosa by aggregates of primarily neutrophils and macrophages surrounded by fewer lymphocytes and plasma cells. The muscularis mucosae is disrupted by the inflammation. Bottom, IHC staining against FCoV in a histology section mirroring the above section. There is extensive positive FCoV cytoplasmic staining for cells at the centre of each aggregate/pyogranuloma in cells with macrophage-like morphology. Scale bars, 250 µm.

virus[6]. These observations are particularly important in view of the *Alphacoronavirus-suis*-related human infections recently observed[19,22].

In 2023, we alerted the veterinary field to an outbreak of FIP in Cyprus, where there had been a concerning increase in cases[11]. Cases were recorded as FIP only if they had compatible clinicopathological signs and a positive test based on quantitative PCR coupled with reverse transcription (RT–qPCR) for FCoV in one of the following samples: peritoneal fluid, pleural fluid, cerebrospinal fluid, fine needle aspiration biopsies or tissue biopsies from granulomatous lesions; or positive FCoV immunohistochemistry (IHC) of granulomatous lesions. Confirmed FIP cases had increased from three and four in 2021 and 2022, respectively, to 215 cases (186 thereof in 2023) from January 2023 to June 2024, representing more than a 50-fold increase. The outbreak emerged in January 2023 in Nicosia, the capital of Cyprus. By February, the area of Nicosia recorded the peak number of cases in any district (Fig. 1a,b). By March, the outbreak had spread to all districts of the Republic of Cyprus observed and was most prevalent in Famagusta (Supplementary Tables 1–6). In June and July, a seeming decline in confirmed cases coincided with a large media awareness campaign to alert veterinarians to the spread of FIP[11], probably leading to most veterinarians diagnosing cases based on clinicopathological findings without performing additional costly confirmatory tests. On 3 August 2023, the Republic of Cyprus minister's cabinet approved the use of human CoV medication stocks in cats with FIP. For access to this medication, among others, a PCR confirmation was required, probably explaining the increased cases recorded during August 2023 (Fig. 1a,b). However, the number of unreported FIP cases in Cyprus remains very high, not least due to the high number of feral cats. Estimates from the Pancyprian Veterinary Association indicate that approximately 10,000 cases of cats presenting with clinical signs of FIP were identified in veterinary clinics alone from January 2023 to July 2023. Furthermore, in October 2023, a first imported case of FIP was confirmed in the UK[34]. The most common clinical form of FIP was effusive (63.72%; Fig. 1c), followed by neurological FIP (26.05%) and the non-effusive (dry) form (10.23%) (Supplementary Table 3). While the initial wave of infections appeared to slow down in August 2023, a second wave can be observed September 2023 to March 2024, with the beginning of a third wave indicated from June 2024. The only distinctive feature of the

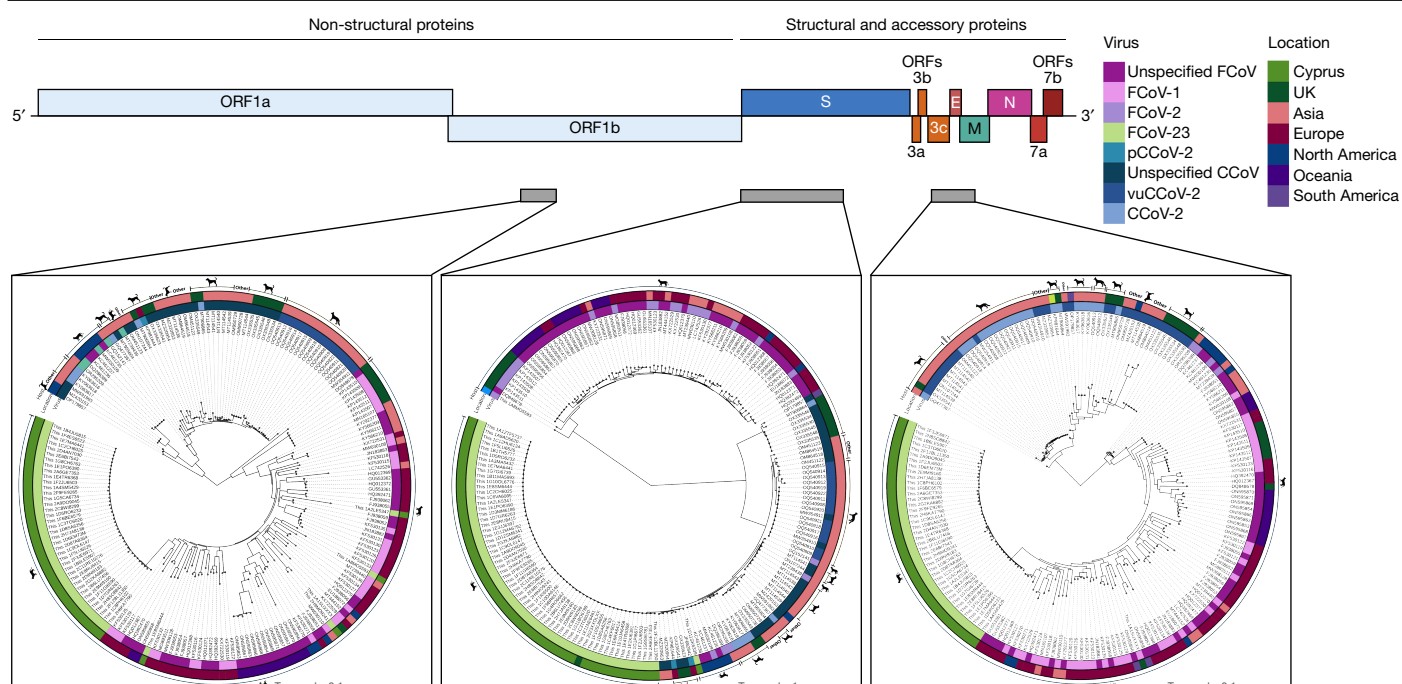

**Fig. 2 | Genomic sequence analysis of Cypriot and UK-import FCoV cases.** Sequences from three different genomic regions (*orf1b*, *S* and *orf3c/E/M*), as indicated by the grey regions marked on the overview of the genome, were obtained through Nanopore sequencing of 45, 63 and 42 Cypriot and UK-import samples, respectively. After initial BLAST analysis, maximum-likelihood trees were generated including other FCoV and CCoV strains to assess the genetic similarity of each region. CCoV-2 genomes are highlighted in blue with pCCoV-2 genomes within this region displayed in a darker blue. FCoV-1 genomes are highlighted in pink and FCoV-2 genomes are shown in purple. Samples from Cyprus (green) and one UK-imported Cypriot cat (dark green) can be seen clustered with FCoV-1 sequences in the *orf1b* and *orf3c/E/M* regions. However, the *S* gene clusters with CCoV-2, most closely with pCCoV-2. Different numbers of sequences are present in each tree due to missing sequence or poor sequence quality and/or alignments in genomes downloaded from NCBI, and due to not all regions being sequenced in all individuals from our samples. Silhouettes representing cats, dogs and foxes were created using BioRender.

second and third waves is a higher proportion of non-effusive forms (Supplementary Tables 3–5). Possible explanations for this include an altered immune response from the host after initial seroconversion and reinfection; that the non-effusive form takes longer to develop; or that cats presenting with the non-effusive form in the early stages of the outbreaks were not diagnosed due to their less common clinical signs.

Where peritoneal or pleural fluid was assessed by cytology, non-degenerative neutrophils admixed with macrophages and small lymphocytes were seen in a protein rich background (Fig. 1d). In total, 17 cases were assessed by histopathology, including intestinal mass (*n* = 8), lymph node (*n* = 5) and kidney (*n* = 4). All showed similar histological features, with multifocal-to-coalescing, pyogranulomatous-to-necrotizing and lymphoplasmacytic inflammation (Fig. 1e). The angiocentric nature can be seen in some areas, while, in others, there is total effacement of the tissue. Immunohistology analysis of FCoV antigen demonstrated a heavy viral load within intralesional macrophages (Fig. 1e).

RNA samples were obtained from 163 confirmed FIP cases between 2021 and November 2023, representing a mixture of geographical origin, sex and clinical presentation (Supplementary Tables 7–12). There are inherent problems with obtaining sufficient read-depth with shotgun sequencing on FIPV RNA samples. The estimates of FCoV sequences versus background (feline) RNA from initial Nanopore-based cDNA sequencing made us opt for a cDNA/PCR-amplification-based Nanopore sequencing to better understand the Cypriot outbreak and to determine whether cat-to-cat transmission is occurring. An initial crude approach based on alignments of publicly available FCoV-1 and FCoV-2 sequences yielded a mixed consensus sequence for the Cypriot outbreak strain, FCoV-23. The consensus was used to design a new primer scheme for targeted amplification of FCoV-23 for Oxford Nanopore-based sequencing. This yielded a total of 20 full-length genomes, 45 partial sequences spanning a section of *orf1b*

(~1,000 bp), 63 spanning the first part of *S* (~2,250 bp) and 42 spanning across *orf3c/E*(ORF4)/*M*(ORF5) (~1,000 bp) (sections indicated in grey in Fig. 2). These include samples from two cats presenting with FIP following recent import from Cyprus to the UK. Other samples were degraded or contained too few viral copies. None of the seven samples from before 2023 amplified (Supplementary Tables 13–18).

The partial *S* region sequence of the Cypriot and UK-import FCoV samples produced two distinct versions of the *S* sequence. BLAST was used to identify close relatives of these *S* sequences. One Cypriot and one UK-import sample are most closely related to an FCoV-1 (MT444152) with 79% similarity. However, the vast majority shows an *S* sequence identified as CCoV-2, flanked by a FCoV-1 sequence. The CCoV-2 sequence is most closely related to the NA/09 strain (JF682842), a hypervirulent pantropic canine coronavirus (pCCoV)[35], at 96.5% sequence identity. Maximum-likelihood analysis found close relationships also to other pCCoV *S* sequences with only partial sequences available (Fig. 2 and Extended Data Fig. 1). This is probably a defining feature of the virus circulating in the outbreak in Cyprus. By contrast, maximum-likelihood trees of regions of *orf1b* and *orf3c/E/M* (Fig. 2) show clustering of all sequences with FCoV-1.

A recombination analysis was carried out comparing the FCoV-23 genome using the representative full-length sequence 2-C11 Re 10276 (GenBank: PQ133182) with FCoV-1, FCoV-2 and CCoV-2 strain CB/05, as NA/09 is not available as a complete genome. Figure 3a visualizes the Bootscan[36] analysis, the RDP5[37] analysis and the pairwise distances between the sequences. Significant *P* values supporting the recombination were reported based on multiple methods, as listed in Supplementary Table 19. The MaxChi[38] breakpoint matrix is shown in Extended Data Fig. 3. Furthermore, a recombination analysis of a domain-0-deletion (ΔD0) FCoV-23 genome 2-F12 BW 1135 is shown in Extended Data Fig. 4. The recombination is very clear and includes a small region of the *orf1b*

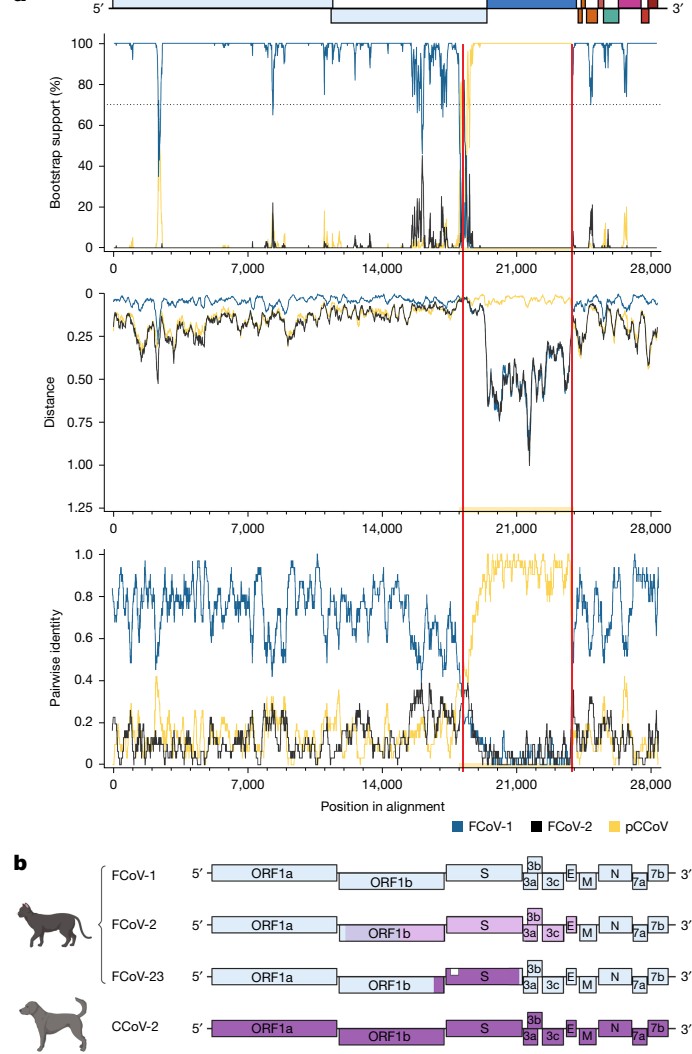

**Fig. 3 | Recombination analysis. a**, Visualizations of a recombination analysis carried out on the assembled FCoV-23 genome 2-C11 Re 10276 (PQ133182) and representative genomes of FCoV-1 (blue), FCoV-2 (black) and pCCoV (yellow). The section highlighted in yellow shows the likely recombination break region, with a red vertical line showing the likely breakpoint. The first panel shows the results of the bootscan analysis; the second panel shows the sequence distance; and the third panel shows the RDP pairwise identity analysis. All three panels show good support for the recombination between FCoV-1 and pCCoV. These results are further supported by a high statistical likelihood of recombination shown in Supplementary Table 21. Recombination analysis of a representative ΔD0 FCoV-23 is shown in Extended Data Figs. 3 and 4. **b**, Schematic of the major recombinations observed in FCoV and CCoV. A common-ancestral-origin virus is thought to have given rise to the original FCoV-1 and CCoV-1 types with further evolution in CCoVs yielding CCoV-2 and CCoV-1/2 serotypes. FCoVs are thought to have recombined with CCoV-2 to form FCoV-2. FCoV-2 and CCoV-2 recombinations have been shown to have different recombination points as highlighted by the different colorations (reviewed previously[23]). The Cypriot FCoV strain, termed FCoV-23, is a recombinant between an FCoV-1 strain and a *S* recombination with a pantropic CCoV-2, pCCoV. Furthermore, deletion variants are observed in the majority of sequenced cases (white box; Fig. 4). The diagram in **b** was created using BioRender.

gene and the majority of *S* with breakpoints around location 18,420 and 23,880. Although there may be a negligible chance of a syndemic—that is, the circulation of two viruses in all of these cats at the same time—we excluded the possibility that this apparent recombination event would reflect only coinfection with two viruses and preferential amplification effects in our multiplex PCR assay by examining the relative positions

of recombination breakpoints and PCR primers. Amplicons 32 and 41 in the tiled amplification scheme span the recombination breakpoints and show clear FCoV|pCCoV and pCCoV|FCoV characteristics in the individual amplicons, respectively. Figure 3b shows the historical breakpoints between FCoV-1 and CCoV-2 that created FCoV-2 alongside the recombination identified in FCoV-23.

Furthermore, Extended Data Fig. 2 shows a neighbour-joining tree for FCoV-23 along with other members of *Alphacoronavirus suis* and a distantly related canine respiratory coronavirus as an outgroup. The assembled genome clusters with representatives of FCoV-1, similar to the clustering of the amplicons outside of *S*.

The main determinant in disease development and transmission of FCoV-23 appears to be the *S* recombination. One of the main suggested determinants of biotype changes, the furin-cleavage site (FCS) at the S1–S2 interface[23,26], is absent in FCoV-2 and also FCoV-23. However, we observed that the majority of samples (>90%) from cats with FIP show the ΔD0 deletion, strongly resembling the deletion observed in TGEV and PRCV (Fig. 4a). While the rest of the genome of FCoV-23 is highly conserved, the ΔD0 deletion appears to be almost cat specific (Fig. 4b). This indicates that this mutation occurs in host rather than being transmitted. However, the only two faecal samples (also affected by sampling bias; <6% of our samples are faeces derived) for which sequences could be obtained show ΔD0. An in-host mutation further raises the question of whether ΔD0 is associated with disease development. Unfortunately, all of our samples so far are from symptomatic, confirmed FIPV cases and further investigation is required to answer this question. Further analysis of ΔD0 was performed by mapping the deletion lengths onto a time-resolved phylogenetic tree using the *orf1ab* and *orf3c/E/M* sequences. The tree shows distinct geographical clades indicating clear transmission patterns of the virus following the epidemiological patterns described in Fig. 1. The spike deletion on the other hand does not show distinct evolutionary patterns supportive of in-cat recombination (Extended Data Fig. 6). Whole-genome alignments confirm the close relationship between all FCoV-23 samples with >99.35% identity over the entire genome in a ClustalW alignment including the different ΔD0 gaps, with two outliers, *E8* and *H9* due to poor sequence quality in these samples (larger gaps; Extended Data Fig. 7). The whole genomes are displayed in the phylogenetic context with the closest related FCoV-2 strain, UU54, pantropic CCoV CB/05 and CCoV-HuPn-2018 (Extended Data Fig. 8).

In other CoVs, including TGEV[39] and CCoV-HuPn-2018[40], D0 was shown to bind to sialosides. Modelling the structure of S against the closely related experimentally confirmed CCoV-HuPn-2018 S[40] shows a much more compact conformation for ΔD0 S and similarity to a structural prediction based on a 'swung out' or a 'proximal' template (Fig. 4c and Extended Data Fig. 5). A number of amino acid changes were observed between 'classical' FECV-2 and FIPV-2 S. In particular, domains A, B and the receptor-binding domain (RBD) show a number of class-changing amino acid changes distinct from FCoV-2 S (Fig. 4a). Modelling the RBD changes against the structure highlights changes at positions 546 and 595, as well as 556, 603 and 636 as being potentially strongly influential to receptor binding properties (Fig. 4d).

Previously indicated key proteins for biotype switch S, ORF3abc and ORF37b were compared to the recently published computational analysis of mutations observed in FECV versus FIPV[26]. The suggested key determinant of FIPV in the FCoV-2 S, position 1404, shows a new amino acid, leucine instead of valine, in FCoV-23. However, this section of spike is derived from FCoV-1. It is therefore unclear how this recombination may impact overall functionality in the context of the rest of S being FCoV-2-like. Other positions in S show a mixture of FIPV versus FECV tendencies (Supplementary Table 23). While a new mutation was identified in ORF3a and ORF3b, no specific indications of pathogenesis could be determined (Supplementary Table 24). Similarly, in ORF7b, two new mutations were identified, but there was no indication of a link to increased pathogenesis (Supplementary Table 25).

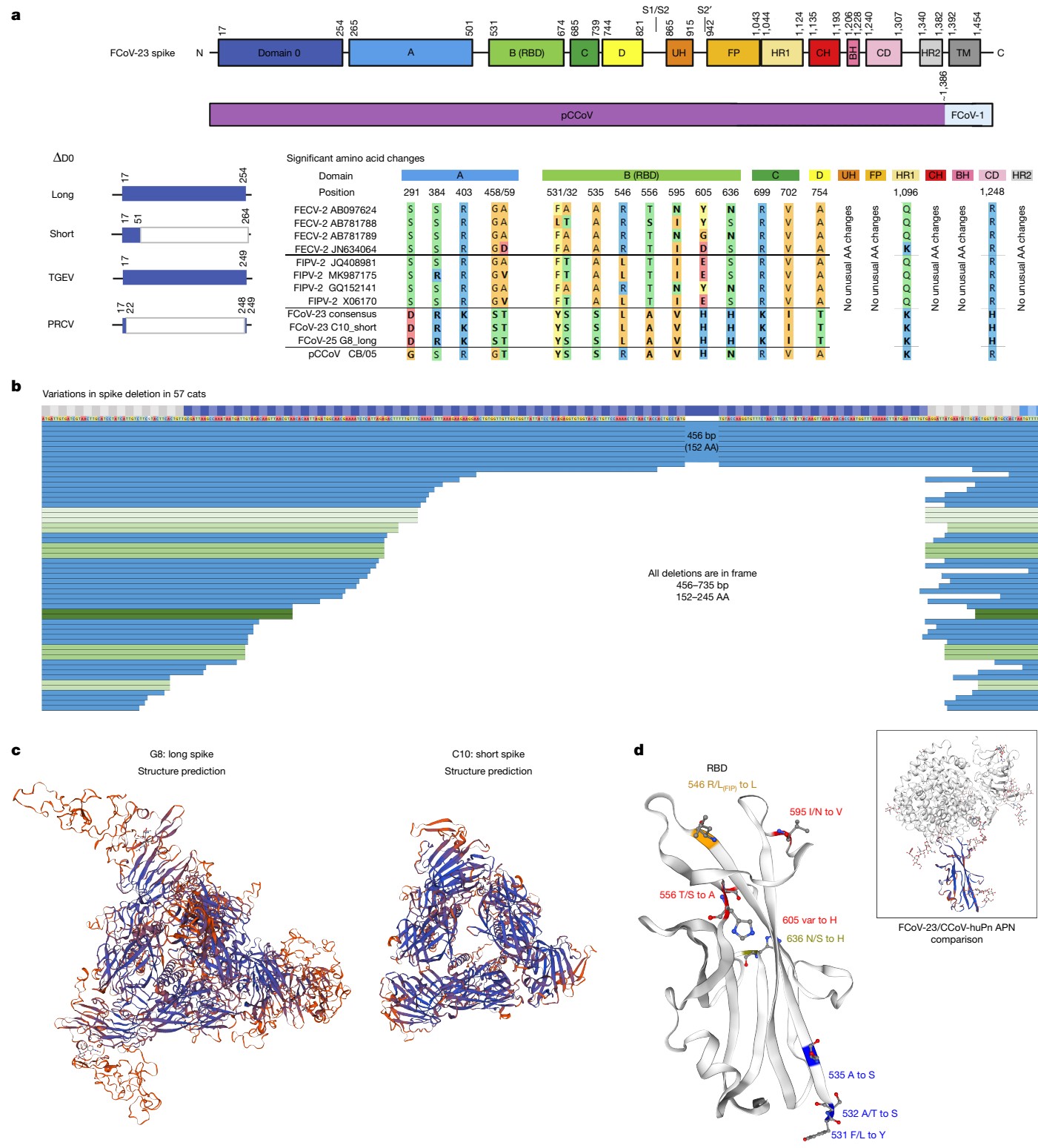

**Fig. 4** | See next page for caption.

## Discussion

Recombination within the *Alphacoronavirus suis* species has been frequently observed previously and has also given rise to the FCoV-2 serotype[5,23,24] (Fig. 3b) in cats. Even complex recombination between FCoV-1 and CCoV has previously been observed[10]. However, here we identified a subtype of FCoV that has recombined with a hypervirulent strain of pCCoV. The recombinant, which we propose naming FCoV-23, shows clearly distinct properties from previously observed FCoV infections. The sequence similarity found between FCoV-23 and CCoV-2 is higher than would be expected without an additional, recent recombination event.

Our data suggest that there is direct transmission of FCoV-23 between cats based on high sequence identity (>99.17% genome-scale including varying *S*-deletion length; Extended Data Figs. 7 and 8), high viral loads (RT–qPCR only) in faeces, as well as the wave-like movement of disease across the island. However, it is unclear whether this virus still needs

**Fig. 4 | Protein sequence and structural analysis of FCoV-23 spike.**
**a**, Analysis of the different domains of FCoV-23 S. D0 shows high similarity to CB/05 pCCoV with a prominent, variable deletion between amino acids (AA) 11–265 resembling the deletion previously observed in TGEV and PRCV S. Multisequence alignment mirroring sequences used for computational analysis previously[26] is shown. Amino acids with side-chain type changes against FCoV-2s, and changes at position 546, where a leucine is more predominant in the FIPV rather than the FECV sequences, are shown. Colours were assigned according to the RasMol amino acid scheme. Amino acid changes that could potentially be associated with biotype change are highlighted in bold. The schematic of spike was created using BioRender. **b**, D0 S deletions in 57 cats (8 full length) are shown from the beginning of S to amino acid 267. Sequences are ordered by the shortest N-terminal sequence. Identical deletions are highlighted in shades of green. All deletions are in-frame. **c**, Structural modelling of the full-length or long S version, represented by sample G8, and the ΔD0 or short version, represented by sample C10. The samples were modelled against CCoV-HuPN-2018, experimentally determined S in the swung-out confirmation 7usa.1.A[40]. The proximal confirmation and comparisons are shown in Extended Data Fig. 8. Colours indicate the confidence, with blue highlighting strong and red highlighting weak confidence. **d**, Modelling of amino acid changes on a structural prediction of the FCoV-23 RBD against CCoV-HuPN-2018 7u8I.1.B. Side-chain type amino acid changes as identified in **a** are highlighted; side chains are shown in blue for variations in FCoV-2s that are distant from the RBD, red for variations that are close to the RBD, orange for the variations at amino acid 546, similar to FIPV-2, and olive for variation differing both from FCoV-2 and pCCoV. A comparison showing a confidence model of the FCoV-23 RBD structure prediction paired with the CCoV-HuPN-2018-canine aminopeptidase N (APN) is shown for orientation and binding visualization at the top right.

a biotype change to result in FIP. A potential reason for differences in pathogenicity may be the acquiring of a deletion of S D0. This is observed in >90% of FIP cases studied here and in a near-host-specific pattern. More work is needed to assess the properties of ΔD0 S, as well as investigation into asymptomatic carriers. While alphacoronaviruses show great cross-neutralizing activity[40], the onset of FIP appears to be rapid and shows little discrimination in the age of the infected cats, highlighting that FCoV-23 is able to circumnavigate pre-existing immunity. If a biotype change is required, it clearly happens far more frequently in FCoV-23 than any of the previously observed FCoV infections. We must therefore study FCoV-23 to gain further understanding of the FECV/FIPV change. The scale of this FIP outbreak has not been observed previously. Concerningly, Cyprus has a high population of unowned cats that are frequently relocated to other parts of Europe, and the wider world. The risk of spreading this outbreak is considerable, as evidenced by the first confirmed UK case[34]. To date (May 2025), we now have observed five confirmed cases of FCoV-23 in the UK: four cats imported from Cyprus and one imported cat from Greece, indicating both local spread from Cyprus to mainland Europe and we also strongly suspect underdiagnosis of FCoV-23 in the UK.

CCoV infections in dogs are typically self-limiting, producing mild enteritis or presenting asymptomatically. Previous work on the FCoV-23 close-relative strains, CCoV NA/09 and CB/05, found that the virus was hypervirulent and infected a range of organs in canine hosts. CCoV typically infects cells of the enteric tract, but pCCoVs were shown to spread to a range of internal organs, including intestine, lung, spleen, liver, kidney, lymph nodes, brain and even T and B cells[35,41–43]. CB/05 has also been identified as responsible for small outbreaks[41].

The clinical signs of FCoV-23 are similar to those observed in classical FIP cases. However, FIP infection itself already shows a strong pantropism, affecting many organs in infected cats[25]. The high number of FCoV-23-positive cells observed in IHC samples is notable (Fig. 1c). This is indicative of high viral loads, but must be validated through quantification. Furthermore, a relatively high percentage (Supplementary Table 4) of confirmed cases in Cyprus presented with neurological signs (26%), which is almost double of what would be expected with classical FIP (14%)[27]. This may be due to increased awareness of presentation or be inherent to FCoV-23 neurotropism; however, further investigation is required.

Substantial changes in the S protein in FCoV-23 may provide some clues as to the enhanced pathogenicity of this virus. FCoV-23, like other FCoV-2s and CCoV-2, does not contain an FCS at the S1/S2, and the related virus CCoV-HuPn-2018 was experimentally found to be uncleaved[40]. Uncleaved S has been shown to be more stable than cleaved S and could enhance faecal shedding and stability in the environment[44]. Conversely, the cleavage at S1/S2 may facilitate the movement of the N-terminal domains and allow the RBD to adopt the receptor-binding-competent form. However, the deletion of D0 may compensate for the increased rigidity. While a similar deletion between TGEV and PRCV was initially indicated as the determinant between a primarily enteric and primarily respiratory tropism, respectively[45,46], a recent study shows a different result in vivo[47]. Sialoside binding is an important feature of CoV infections and may contribute to intracellular spread[48]. It is therefore surprising that FCoV-23 loses the sialoside-binding D0. However, while binding to sialosides can enhance virus attachment and entry into host cells, binding to sugars can lead to increased retention of virus on producer cells. While some viruses, like influenza, solve this issue by encoding their own sugar-cleaving enzyme, neuraminidase, CoVs must find a balance through modifying glycoprotein binding[49]. Losing D0 could therefore be a trade-off to enhance virus release. This may be compensated by the enhanced flexibility of S and increased binding efficiency at the RBD through mutation. These are manifold in the FCoV-23 pCCoV recombinant and in particular mutations at position 546 (already possibly associated with FIP) and 595 are likely to have a strong impact on aminopeptidase N receptor binding (Fig. 4c).

Recombination between FCoV-1 and pCCoVs is not surprising in that (1) such recombinations have been observed previously and (2) pCCoVs have evolved in Greece and south eastern Europe[35,42]. However, the emergence of pCCoV in Greece happened over 1 year ago, so the question as to why this happened now is unclear. One possible explanation is the 'right mutation, right time, right place' theory. Recombination between feline and canine CoVs happen frequently but, as previous reports show, they rarely spread. Introduction of a more successful, spreading variant to a dense population, like the cats in Cyprus, may be sufficient to allow this virus to cause an outbreak. However, the acquisition of a likely in-host deletion adds a further level of concern to CoV evolution. These viruses already show a mutation rate similar to influenza viruses (despite proofreading)[50], high prevalence for recombination[51] and, here, we showed that they are able to acquire substantial in-host insertion–deletion mutations that may completely change their cell binding properties and tropism.

This paper reports the emergence of a new *Alphacoronavirus suis* feline/canine coronavirus that shows high spreading potential with the associated pathology of lethal FIP if left untreated. Further investigations into the properties of this virus are now essential. While antiviral agents including GS-441524 and molnupiravir were successfully used to treat many cats affected by FCoV-23, early treatment is essential but the associated costs can be prohibitive. Prevention of spread and the development of vaccines are important to stop this epizootic virus from spreading beyond Cyprus.

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

## Methods

### Enrolment of FIP cases in Cyprus

The electronic records of Vet Dia Gnosis in Limassol, Cyprus (at present, the only Veterinary diagnostics laboratory on the island), were searched for any positive FCoV RT–qPCR or IHC cases from September 2021 up to June 2024. The cases to be enrolled needed to have compatible clinicopathological findings for FIP as recently outlined by the European Advisory Board on Cat Diseases Guidelines[25] as well having a positive FCoV IHC (on tissue biopsies with granulomatous lesions) or a positive RT–PCR for FCoV in one of the following samples: peritoneal fluid, pleural fluid, cerebrospinal fluid and granuloma fine needle aspiration biopsies or tissue biopsies. The original samples were submitted to the Vet Dia Gnosis commercial laboratory (Limassol, Cyprus) by local veterinarians and were then submitted to the Laboklin commercial laboratory. Ethical approval for this study was granted by the Pancyprian Vet Association. According to the terms and conditions of the Vet Dia Gnosis, as well as the Cypriot legislation (The Dogs LAW, N. 184 (I)/2002), no special permission from animal owners or the animal welfare commission is needed for additional testing on residual sample material once diagnostics are completed. According to the terms and conditions of the Laboklin laboratory, as well as the RUF-55.2.2.2532-1-86-5 decision of the government of Lower Franconia, no special permission from animal owners or the animal welfare commission is needed for additional testing on residual sample material once diagnostics are completed. The study was also approved by the Veterinary Ethical Review Committee, The Royal (Dick) School of Veterinary Studies (R(D) SVS), The University of Edinburgh (VERC reference: 233.23).

### Sample from cats imported to the UK

Two UK veterinary practices contacted the R(D)SVS with suspected cases of FIP in cats recently imported from Cyprus. Peritoneal fluid samples were taken from the cats and sent to R(D)SVS for further testing and for sequencing at The Roslin Institute with the consent of the cats' owners. The veterinary practices and the cats' owners were kept informed at all stages.

### Histopathology and IHC

A subset of cases ($n$ = 4) from 17 non-effusive (dry) FIP cats diagnosed after January 2023 with available tissue specimens was selected. Tissue specimens were carefully obtained and immediately fixed in 10% buffered formalin. After fixation, the tissues were processed by embedding, 4 μm sectioning and subsequent staining with haematoxylin and eosin (H&E). The inclusion criteria were defined based on comprehensive histopathological assessments. Specifically, emphasis was placed on identifying the characteristic FIP-associated histological hallmarks, which encompassed vasculitis, phlebitis and periphlebitis. Consecutive tissue sections were mounted onto charged slides. After pretreatment for 5 min at 110 °C in 0.01 M pH 6 citrate buffer, the slides were incubated for 30 min at room temperature with primary mouse monoclonal anti-feline coronavirus antibodies at 1:400 (Bio-Rad, MCA 2194). The EnVision anti-mouse system was used for visualization according to the manufacturer's instructions (Agilent).

### RNA extraction and cDNA synthesis

RNA from specimens from Cyprus underwent automated total nucleic acid extraction using the MagNA Pure 96 DNA AND Viral NA Small Volume kit (Roche Diagnostics). RT–PCR for FCoV was performed at Laboklin[52].

RNA samples from cats imported to the UK from Cyprus were extracted using the QIAamp viral RNA extraction kit (Qiagen) according to the manufacturer's instructions.

cDNA synthesis for all RNA samples was performed using LunaScript RT SuperMix Kit (NEB) with 16 μl template RNA in 20 μl reactions according to the manufacturer's instructions.

### Primer design and PCR amplification

A set of primers for 1 kb/100 bp overlap tiled amplification were initially designed with available FCoV genomes on NCBI[53] using primalscheme[54], and by manual redesign (Supplementary Table 19). These were used to generate a composite representative FCoV-23 genome. Using this genome, a new, 800 bp/80 bp overlap primer scheme was generated to sequence FCoV-23. A list of the primer sequences used here is provided in Supplementary Table 20. The PCR reactions for these multiplexed amplifications were performed using the same conditions as described below, but with multiplexed primers.

For a subset of samples, DNA was amplified using primers only targeting parts of spike, *orf1b* and *orf3c/E/M*. A list of these primers is provided in the first four rows of Supplementary Table 19.

cDNA synthesis was performed using the LunaScript reverse transcriptase supermix (NEB) followed by PCR amplification with Q5 (NEB). We used 1.25 μl 10 μM forward and reverse primers and 3 μl cDNA in a 25 μl reaction. Amplification was performed under the following PCR conditions: initial denaturation at 98 °C for 30 s, 40 cycles of denaturation at 98 °C for 10 s, annealing and extension at 65 °C for 4 min, followed by a final extension at 65 °C for 5 min. Amplified DNA was purified using AMPure XP beads (Beckman Coulter).

### Nanopore sequencing

Amplicons were quantified using Qubit (Thermo Fisher Scientific) high-sensitivity assays and diluted to 150 ng DNA per sample in 12 μl nuclease-free water. Libraries were prepared using the Oxford Nanopore Technologies' (ONT) NBD112.96 ligation kit according to the manufacturer's protocol for amplicon sequencing with some modifications. Owing to an unavailability of the NBD112.96 kit reagent AMII H, after ligation of barcodes, a second end-prep was performed using the Ultra II End Repair module (NEB). The rest of the protocol was carried out from the adapter ligation stage according to the manufacturer's protocol for LSK112 using the AMX-F adapter supplied in the early-access Q20+ version of kit 112. The library was loaded onto the MinION R10.4 flow cell and sequencing was performed on a GridION sequencing device.

To identify the virus present in one of the UK cases as quickly as possible, only the spike amplicon was amplified and the sample was sequenced using the ONT RAD004 rapid sequencing kit on an R9.4 flow cell according to the manufacturer's protocol. The other UK case was amplified at an earlier time and was included in a sequencing run with the Cypriot samples; the data from that sample were treated in the same manner as the Cypriot samples, and that sample was later found to have the non-recombinant FCoV-1 spike.

### Bioinformatic analysis

Basecalling and demultiplexing was performed on the GridION sequencing device (ONT) using Guppy (v.2.24-r1122) in super-accurate mode, specifying --require_barcodes_both_ends and using the appropriate super-accurate basecalling models for each of the different sequencing methods used. For the spike amplicons, after basecalling, amplicon_sorter[55] (v.2023-06-19) was used to identify consensus amplicons between 3 kb and 5 kb without using a reference. Spike amplicons were identified from the output through alignment with minimap2[56] (v.2.22) to the spike from the first sample we sequenced (1-G7_Gi_6739) for which a consensus was made using the same process, with the correct amplicon identified by BLAST[57]. The identified amplicons were polished with medaka (v.1.8.0, ONT). For the UK case that was sequenced using the rapid kit, because the reads were fragmented by the library preparation process, reads were used to polish the 1-G7_Gi_6739 sequence using medaka. The reads were aligned to the polished sequence with minimap2 and visualized using IGV[58] (v.2.11.1) to visually confirm the reads supported the consensus sequence and confirm that it had not been biased by the reference used.

The amplicons in POL1ab and ORF3c/E/M were assembled and polished using LILO[59]. Multisequence alignments were carried out for each of the regions using mafft[60] (v.7.49) against all complete genomes of CCoV and FCoV genomes available on NCBI using the --adjustdirection flag. Alignments were visualized in Mega7[61]. All downloaded genomes were trimmed down to each of the target regions amplified from our samples. Maximum-likelihood trees were constructed using IQ-TREE[62] (v.2.0.5). TempEst[63] (v.1.5.3) was used to determine the best-fitting root for the trees, and visualizations and annotations of the trees were done using iTOL[64] (v.6.8.1).

To assemble a representative genome for the population, the combined reads of several samples for which we had sequencing data were run through LILO. The polished amplicons were mapped to a single fasta file containing an FCoV-1 (MT239440.1) and a CCoV-2 (KP981644.1) using minimap2 and visualized with IGV, with only the amplicons from the spike region aligning to CCoV-2. Representative amplicons covering the entire genome were selected and scaffolded using scaffold_builder[65] with MT239440.1 as a reference. The raw reads were trimmed by 50 bp to remove adapters and barcodes and they were used to polish this scaffold using medaka. Mafft was used to align this genome to representative genomes from *Alphacoronavirus 1* with canine respiratory coronavirus as an outgroup, and Mega7 was used to create a neighbour join tree, which was visualized and annotated with iTOL.

Having assembled a representative genome, primers were redesigned to include ambiguous bases that supported the sequence in the representative genome, and these are the primers that were used to amplify 800 bp tiled amplicons for individual Cypriot samples. The Cypriot samples were assembled using LILO, with the representative genome as the reference. Assembled genomes had their reads aligned back to them using minimap2 and were visualized with IGV. Each genome was visually inspected for errors and corrections made, primarily where primer sequences had been incorrectly incorporated. Where there were gaps in these genomes, and there were reads that spanned the gap but did not reach the depth threshold for LILO, reads were extracted and amplicon_sorter was used to attempt to create a consensus for the missing amplicon. These amplicons were carefully visually inspected for errors before being incorporated into the assembly to close gaps using scaffold_builder. Multisequence alignments between the population level representative genome. a sample without a spike D0 deletion (2-C11_Re_10276, PQ133182) and a sample with a D0 deletion (2-F12_Bl_11350, PQ133177) against MT239440.1, LC742526.1 and KP981644.1 was carried out using ClustalW[66] in Mega7, and recombination analyses carried out using RDP5[37] (v.5.45) with the default settings.

A list of all (near) complete genome sequences generated in this study is provided in Supplementary Table 26.

### Structure prediction of spike deletion variants

SWISS-MODEL structure prediction and analysis (https://swissmodel.expasy.org/, accessed October 2023)[67] were used to model the partial, high-confidence N-terminal sequence of a full-length spike (1008 amino acids) and a deletion spike (797 amino acids). To model the G8 Cypriot full-length spike, we used the *Alphacoronavirus 1* experimentally resolved CCoV-HuPn-2018[40] structure as a template. Modelling against the 7usa.1.a, swung out confirmation, yielded a GMQE of 0.60 and a global QMEAND of 0.68 ± 0.05, with a sequence identity of 88.97%; and against 7us6.1.A, proximal confirmation, yielded a GMQE of 0.62 and global QMEAND of 0.58 ± 0.05, with a sequence identity of 81.03%. Modelling of C10 spike, C-terminal deletion, against 7usa.1.A, swung out confirmation, yielded a GMQE of 0.74 and a hlobal QMEAND of 0.69 ± 0.05, with a sequence identity of 88.66%; and against 7us6.1.A, proximal confirmation of HuPN, yielded a GMQE of 0.76 and a global QMEAND of 0.70 ± 0.05, with a sequence identity of 88.42%.

The RBD was modelled using the 7u0I.1B structure, again CCoV-HuPn-2018[40] complexed with canine APN as a template. The GMQE was 0.89 and the global QMEAND was 0.85 ± 0.07, with a sequence identity of 92.41%. PDB files are available reposited at Zenodo[68] (https://doi.org/10.5281/zenodo.15488938).

### Reporting summary

Further information on research design is available in the Nature Portfolio Reporting Summary linked to this article.

## Data availability

Full genome sequences of FCoV-23 are reposited in GenBank accession numbers PQ133176–PQ133195 and listed in full in Supplementary Table 26. Sequence alignment files underlying Fig. 2, protein structure prediction files underlying Fig. 4, case numbers underlying Fig. 1, as well as another copy of the whole-genome sequence files are available at Zenodo[68] (https://doi.org/10.5281/zenodo.15488938).

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

**Acknowledgements** We thank the members of the Pancyprian Veterinary Association and all of the veterinarians in Cyprus for sample submission. This work was supported by EveryCat Health Foundation award number EC23-OC1 (C.A., D.G.-M., C.T.-B.), Morris Animal Foundation/Every Cat Foundation 'FCoV-23 Centre of Excellence' award numbers DE25FE-705 and DE25FE-705 (C.T.-B.), Vet Dia Gnosis (C.A.), as well as the BBSRC Institute Strategic Programme grant funding to the Roslin Institute BBS/E/RL/230002C (A.S.W., C.T.-B.) and BBS/E/RL/230002D (S.J.L., S.M.), and grant numbers BBS/E/D/20241866, BBS/E/D/20002172 and BBS/E/D/20002174 (C.T.-B.). A.S.W. is a BBSRC discovery fellow on grant number BB/W009870/1.

**Author contributions** Conceptualization: C.A., A.S.W., D.G.-M., S.M. and C.T.-B. Methodology: C.A., A.S.W., S.M. and C.T.-B. Validation: C.A., A.S.W., S.M. and C.T.-B. Formal analysis: A.S.W., S.M. and C.T.-B. Investigation: C.A., A.S.W., D.E., A.J.H., M.O., S.F., A.M., M.L., R.H. and C.T.-B. Resources: C.A., A.S.W., D.E., M.L., A.H., A.Z., S.L., M.G. and C.T.-B. Data curation: A.S.W., S.M. and C.T.-B. Writing—original draft preparation: C.A., A.S.W. and C.T.-B. Writing—review and

editing: C.A., A.S.W., S.F., D.G.-M., S.J.L., S.M. and C.T.-B. Visualization: A.S.W., S.J.L., S.M. and C.T.-B. Supervision: C.A. and C.T.-B. Project administration: C.A. and C.T.-B. Funding acquisition: C.A. and C.T.-B.

**Competing interests** Laboklin is a veterinary laboratory offering diagnostic services, including bacteriological and molecular biological examinations. Vet Dia Gnosis offers veterinary pathology diagnostic services only. M.G. is employed by Laboklin. C.A. and S.L. are the co-founders of Vet Dia Gnosis. C.A. is an external collaborator of Vet Dia Gnosis and S.L. and A.Z. are employed by Vet Dia Gnosis.

**Additional information**
**Correspondence and requests for materials** should be addressed to Charalampos Attipa, Amanda S. Warr or Christine Tait-Burkard.

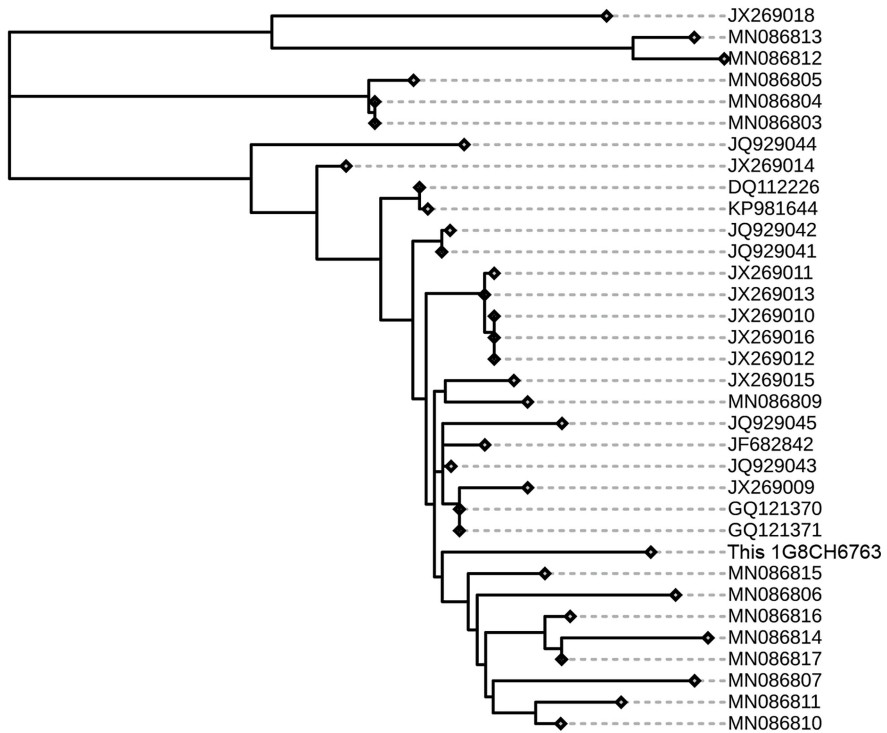

**Extended Data Fig. 1 | Relationship to other pCCoV viruses.** Maximum likelihood tree of pantropic CCoV spikes (~450 bp region). Alignments were done with Muscle in MEGA7, and maximum likelihood tree was made in MEGA7 with default settings. Tree was visualized in iTOL. Extended Data Fig. 1 shows a maximum likelihood tree generated from the alignment of known pCCoV spike amplicons with the non-deletion form of the FCoV-23 amplicon. The alignment was carried out with Muscle in MEGA7 and the maximum likelihood tree was generated with MEGA7 on default settings. The tree was visualized with iTOL. The region targeted is ~450 bp with longer sequences trimmed down. The FCoV-23 spike with the deletion could not be used as the deleted region heavily overlaps with the region in the alignment. The FCoV-23 amplicon clusters among the pCCoV sequences.

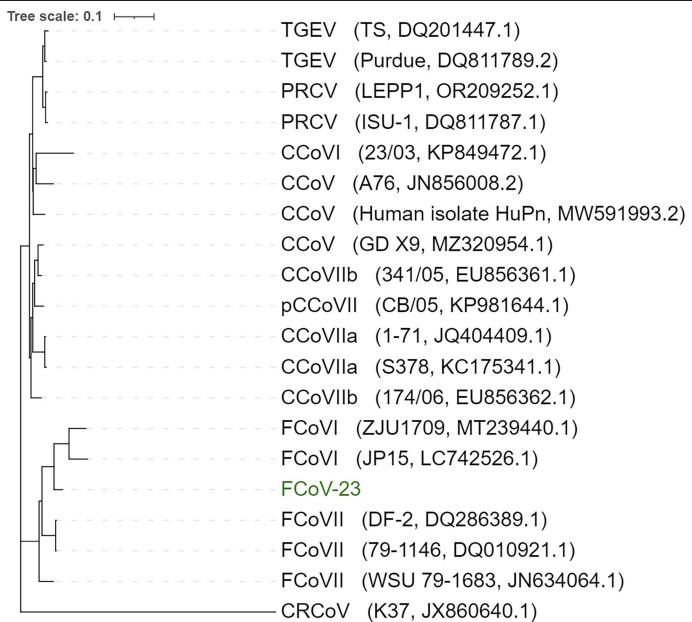

Tree scale: 0.1 ⊢━━━┥

TGEV    (TS, DQ201447.1)
TGEV    (Purdue, DQ811789.2)
PRCV    (LEPP1, OR209252.1)
PRCV    (ISU-1, DQ811787.1)
CCoVI    (23/03, KP849472.1)
CCoV    (A76, JN856008.2)
CCoV    (Human isolate HuPn, MW591993.2)
CCoV    (GD X9, MZ320954.1)
CCoVIIb    (341/05, EU856361.1)
pCCoVII    (CB/05, KP981644.1)
CCoVIIa    (1-71, JQ404409.1)
CCoVIIa    (S378, KC175341.1)
CCoVIIb    (174/06, EU856362.1)
FCoVI    (ZJU1709, MT239440.1)
FCoVI    (JP15, LC742526.1)
FCoV-23
FCoVII    (DF-2, DQ286389.1)
FCoVII    (79-1146, DQ010921.1)
FCoVII    (WSU 79-1683, JN634064.1)
CRCoV    (K37, JX860640.1)

**Extended Data Fig. 2** | Maximum likelihood tree of FCoV-23 with other alphacoronavirus 1 full genome sequences.

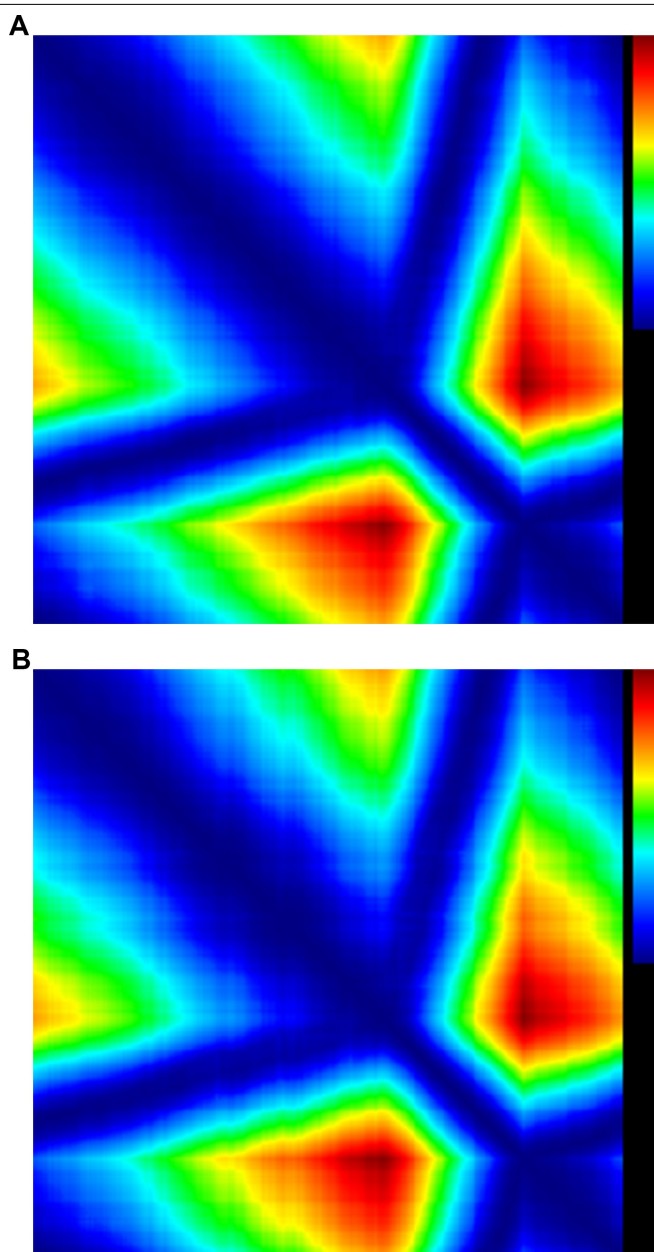

**Extended Data Fig. 3 | MaxChi breakpoint matrix from RDP5 analysis for A) spike full-lenth FCoV-23 genome 2-C11 Re 10276 and B) domain 0-deletion FCoV-23 genome 2-F12 BW 11350.** MaxChi breakpoint matrix generated with default settings in RDP5. Extended Data Fig. 3 shows the MaxChi7 breakpoint matrix generated in RDP58. The dark red region highlights the likely recombination breakpoints, which align with the breakpoints described in the main text.

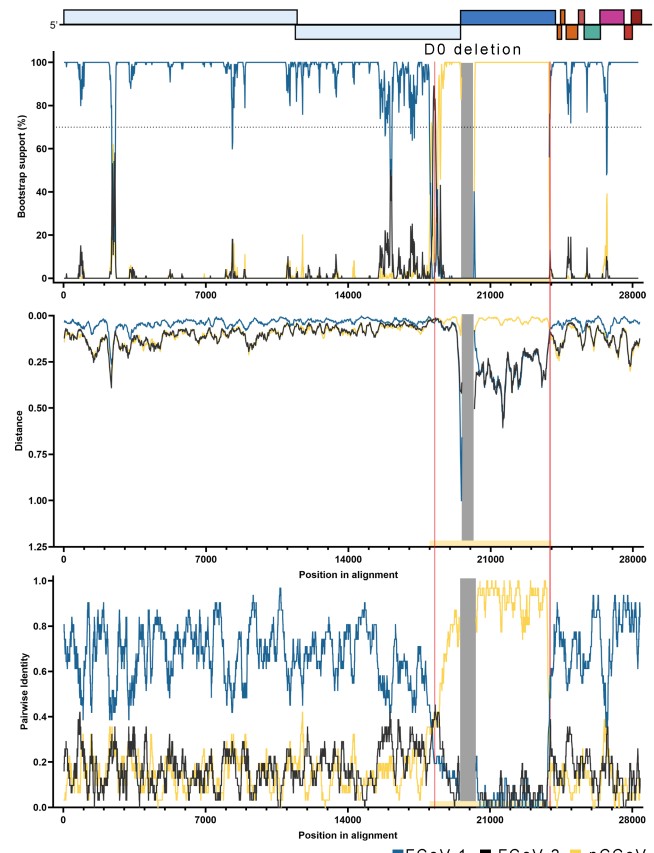

**Extended Data Fig. 4 | Recombination analysis of domain 0-deletion FCoV-23 genome 2-F12 BW 11350.** A) Visualizations of a recombination analysis carried out on the assembled FCoV-23 genome and representative genomes of FCoV-1 (blue), FCoV-2 (black) and pCCoV (yellow). The yellow panel shows the likely recombination break region, with a red vertical line showing the likely break point. The first panel shows the results of the Bootscan analysis, the second panel shows the sequence distance, and the third panel shows the RDP pairwise identity analysis. All three panels show good support for the recombination between FCoV-1 and pCCoV. These results are further supported by high statistical likelihood of recombination shown in Supplementary Table 22.

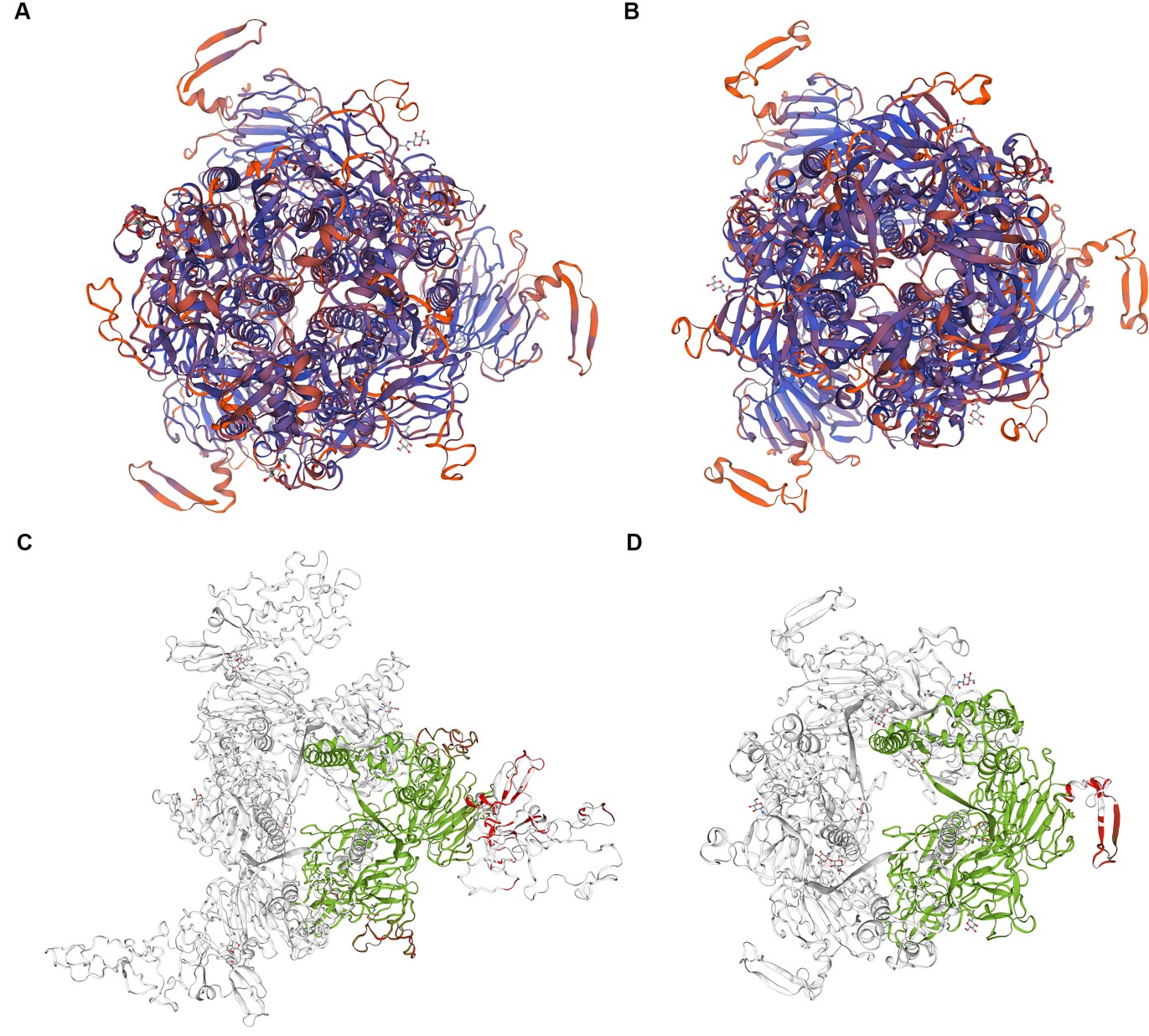

**Extended Data Fig. 5 | Protein structure modelling.** Structural modelling of A) G8 full-length and B) C10 domain 0 deletion (short) spike using the 7us6.1.A proximal confirmation ofCCoV-HuPN-2018 as a template. C&D) Comparison between the G8 full-length and the C10 domain 0 deletion (short) spike structure prediction. C) represents the swung out (modelled against the 7usa.1.A template) and D) the proximal confirmation (modelled against the 7us6.1.A template). Colors represent consistency with red being inconsistent and green consistent between the two structures.

# 1ab+3em with Spike deletion

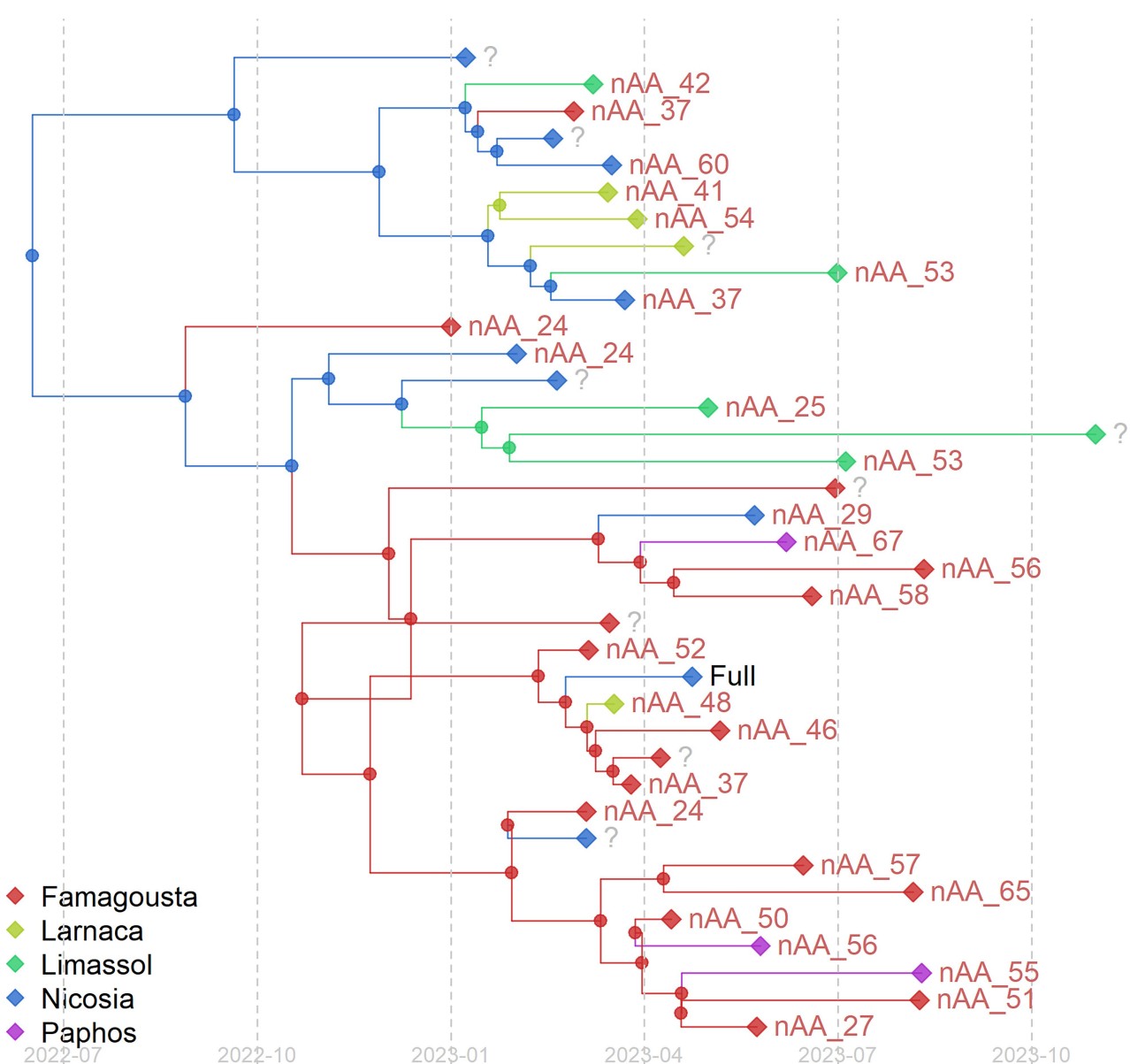

**Extended Data Fig. 6 | Bayesian time-resolved phylogenetic trees for orf1ab and orf3-E-M with annotation of correlated spike D0 deletion lengths.** Origin of the virus sample in Cyprus are given different colors as indicated in the legend. Sequences of the FCoV "backbone" orf1ab and orf3-E-M regions fall somewhat into regional clades, with the inferred epidemic origin in Nicosia, and subsequent transmissions into most other regions (Famagousta, Larnaca, and Limassol), although transmissions into Paphos are inferred to come from Famagousa. The tips are labelled according to how many amino acids remain in affected genomic region of spike, these range from nAA_24 (24 amino acids) to Full (270 amino acids). Orf1ab+3em sequences with unknown spike length are denoted with "?". The number of amino acids remaining does not appear to show an evolutionary pattern, for example the figure does not show that the number of amino acids decreases over time within a clade. Instead, the figure shows a more random pattern of spike deletions, which would be indicative of individually occurring within-host processes, such as in-cat recombination.

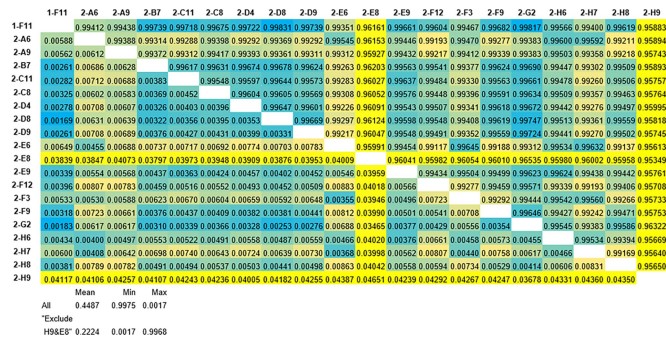

**Extended Data Fig. 7 | Sequence identity/divergence of FCoV-23 full genome sequences.** The number of base substitutions/identical bases per site from between sequences are shown. Analyses were conducted using the Maximum Composite Likelihood model using 20 full-genome FCoV-23 sequences using 29,144 base positions. All Codon positions (1/2/3 + NC) were included. Ambiguous positions were removed using the pairwise deletion option. Evolutionary analyses were conducted in MEGA X and are displayed (bottom) with or without sequences H9 and E8, which display poorer sequence quality (lower coverage and bigger gaps) and should be treated with caution.

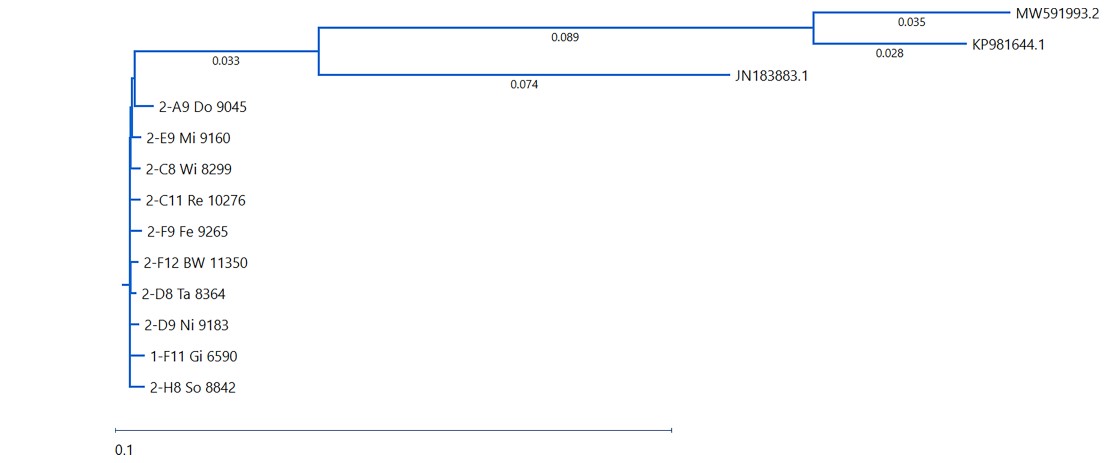

**Extended Data Fig. 8 | Clustal W whole genome alignment.** 10 gap-free FCoV-23 genome sequences were multi-genome aligned using ClustalW with JN183883 (closest FCoV-2, UU54), KP981644 (pantropic CCoV CB/05), and MW591993 (CCoV-HuPn-2018) for context.

# Reporting Summary

Please do not complete any field with "not applicable" or n/a. Refer to the help text for what text to use if an item is not relevant to your study.
For final submission: please carefully check your responses for accuracy; you will not be able to make changes later.

## Statistics

For all statistical analyses, confirm that the following items are present in the figure legend, table legend, main text, or Methods section.

| n/a | Confirmed | |
|---|---|---|
| ☐ | ☑ | The exact sample size (*n*) for each experimental group/condition, given as a discrete number and unit of measurement |
| ☐ | ☑ | A statement on whether measurements were taken from distinct samples or whether the same sample was measured repeatedly |
| ☒ | ☐ | The statistical test(s) used AND whether they are one- or two-sided<br>*Only common tests should be described solely by name; describe more complex techniques in the Methods section.* |
| ☒ | ☐ | A description of all covariates tested |
| ☒ | ☐ | A description of any assumptions or corrections, such as tests of normality and adjustment for multiple comparisons |
| ☐ | ☑ | A full description of the statistical parameters including central tendency (e.g. means) or other basic estimates (e.g. regression coefficient) AND variation (e.g. standard deviation) or associated estimates of uncertainty (e.g. confidence intervals) |
| ☒ | ☐ | For null hypothesis testing, the test statistic (e.g. *F*, *t*, *r*) with confidence intervals, effect sizes, degrees of freedom and *P* value noted<br>*Give P values as exact values whenever suitable.* |
| ☐ | ☑ | For Bayesian analysis, information on the choice of priors and Markov chain Monte Carlo settings |
| ☒ | ☐ | For hierarchical and complex designs, identification of the appropriate level for tests and full reporting of outcomes |
| ☒ | ☐ | Estimates of effect sizes (e.g. Cohen's *d*, Pearson's *r*), indicating how they were calculated |

*Our web collection on statistics for biologists contains articles on many of the points above.*

## Software and code

Policy information about availability of computer code

| Data collection | Guppy (v2.24-r1122), medaka (v1.8.0), minimap2, LILO, scaffold_builder, R Statistical Software (R-4.5.0). |
|---|---|
| Data analysis | GraphPad Prism 10.4.2, iTOL (v6.8.1), IGV (v2.11.1), BLAST, mafft (v7.49), IQ-Tree (v2.0.5, TempEST (v1.5.3), Mega7, RDP5 (v.5.45), SWISS-MODEL |

For manuscripts utilizing custom algorithms or software that are central to the research but not yet described in published literature, software must be made available to editors and reviewers. We strongly encourage code deposition in a community repository (e.g. GitHub). See the Nature Portfolio guidelines for submitting code & software for further information.

## Data

Policy information about availability of data

All manuscripts must include a data availability statement. This statement should provide the following information, where applicable:
- Accession codes, unique identifiers, or web links for publicly available datasets
- A description of any restrictions on data availability
- For clinical datasets or third party data, please ensure that the statement adheres to our policy

Full genome sequences of FCoV-23 are reposited in Genbank accession numbers PQ133176-133195 and listed in full in Supplementary table 26. Sequence alignment files underlying Figure 2, protein structure prediction files underlying Figure 4, case numbers underlying Figure 1, as well as another copy of the whole genome sequence files are available at Zenodo (https://doi.org/10.5281/zenodo.15488938).

## Research involving human participants, their data, or biological material

Policy information about studies with human participants or human data. See also policy information about sex, gender (identity/presentation), and sexual orientation and race, ethnicity and racism.

| | |
|---|---|
| Reporting on sex and gender | N/A |
| Reporting on race, ethnicity, or other socially relevant groupings | N/A |
| Population characteristics | N/A |
| Recruitment | N/A |
| Ethics oversight | N/A |

Note that full information on the approval of the study protocol must also be provided in the manuscript.

## Field-specific reporting

Please select the one below that is the best fit for your research. If you are not sure, read the appropriate sections before making your selection.

☑ Life sciences        ☐ Behavioural & social sciences        ☐ Ecological, evolutionary & environmental sciences

For a reference copy of the document with all sections, see nature.com/documents/nr-reporting-summary-flat.pdf

## Life sciences study design

All studies must disclose on these points even when the disclosure is negative.

| | |
|---|---|
| Sample size | |
| Data exclusions | |
| Replication | |
| Randomization | |
| Blinding | |

## Behavioural & social sciences study design

All studies must disclose on these points even when the disclosure is negative.

| | |
|---|---|
| Study description | |
| Research sample | |
| Sampling strategy | |
| Data collection | |
| Timing | |
| Data exclusions | |
| Non-participation | |
| Randomization | |

# Ecological, evolutionary & environmental sciences study design

All studies must disclose on these points even when the disclosure is negative.

| | |
|---|---|
| Study description | Outbreak study of feline infectious peritonitis in Cyprus |
| Research sample | Clinically and RT-qPCR- or IHC-confirmed, physiological fluids from cats |
| Sampling strategy | All cases with confirmed clinical and RT-qPCR or IHC history. |
| Data collection | |
| Timing and spatial scale | Oct 2022 to Jun 2024 |
| Data exclusions | N/A - inclusion criteria rather than exclusion |
| Reproducibility | |
| Randomization | N/A - no intentional randomization |
| Blinding | Double blind |

Did the study involve field work?  ☑ Yes  ☐ No

## Field work, collection and transport

| | |
|---|---|
| Field conditions | Cypriot veterinary practitionners |
| Location | Cyprus / Middle East / Southeastern Europe |
| Access & import/export | Good access. Import and export of samples possible |
| Disturbance | We are still in the process of trying to acquire samples from surrounding countries. Indications are that the outbreak in fact started in the Turkish part of Cyprus but until next year March no discussions on neutral ground can be had. Similar, by rumours affected areas include the south of Lebanon and possibly Israel but due to the war, no access is possible. |

# Reporting for specific materials, systems and methods

We require information from authors about some types of materials, experimental systems and methods used in many studies. Here, indicate whether each material, system or method listed is relevant to your study. If you are not sure if a list item applies to your research, read the appropriate section before selecting a response.

### Materials & experimental systems

| n/a | Involved in the study |
|---|---|
| ☐ | ☑ Antibodies |
| ☒ | ☐ Eukaryotic cell lines |
| ☒ | ☐ Palaeontology and archaeology |
| ☐ | ☑ Animals and other organisms |
| ☒ | ☐ Clinical data |
| ☒ | ☐ Dual use research of concern |
| ☒ | ☐ Plants |

### Methods

| n/a | Involved in the study |
|---|---|
| ☒ | ☐ ChIP-seq |
| ☒ | ☐ Flow cytometry |
| ☒ | ☐ MRI-based neuroimaging |

## Antibodies

| | |
|---|---|
| Antibodies used | BioRad, MCA 2194 |
| Validation | Supplier validated as well as known positive samples. |

# Eukaryotic cell lines

Policy information about cell lines and Sex and Gender in Research

| | |
|---|---|
| Cell line source(s) | |
| Authentication | |
| Mycoplasma contamination | |
| Commonly misidentified lines (See ICLAC register) | |

# Palaeontology and Archaeology

| | |
|---|---|
| Specimen provenance | |
| Specimen deposition | |
| Dating methods | |

☐ Tick this box to confirm that the raw and calibrated dates are available in the paper or in Supplementary Information.

| | |
|---|---|
| Ethics oversight | |

Note that full information on the approval of the study protocol must also be provided in the manuscript.

# Animals and other research organisms

Policy information about studies involving animals; ARRIVE guidelines recommended for reporting animal research, and Sex and Gender in Research

| | |
|---|---|
| Laboratory animals | No laboratory animals were used in this study |
| Wild animals | No wild animals were used in this study |
| Reporting on sex | In the supplementary tables |
| Field-collected samples | Samples were collected by veterinarians upon clinical examination for diagnostic purposes from cats presented by their owners or guardians to investigate and treat FIP. Samples used in this study were leftovers from diagnostic samples only. |
| Ethics oversight | Ethical approval for this study was granted by the Pancyprian Vet Association. According to the terms and conditions of the Vet Dia Gnosis, as well as the Cypriot legislation [The Dogs LAW, N. 184 (I)/2002], no special permission from animal owners or the animal welfare commission is needed for additional testing on residual sample material once diagnostics are completed. According to the terms and conditions of the LABOKLIN laboratory, as well as the RUF-55.2.2.2532-1-86-5 decision of the government of Lower Franconia, no special permission from animal owners or the animal welfare commission is needed for additional testing on residual sample material once diagnostics are completed. The study was also approved by the Veterinary Ethical Review Committee, The Royal (Dick) School of Veterinary Studies (R(D)SVS), The University of Edinburgh, UK (VERC Reference: 233.23). |

Note that full information on the

# Clinical data

Policy information about clinical studies

All manuscripts should comply with the ICMJE guidelines for publication of clinical research and a completed CONSORT checklist must be included with all submissions.

| | |
|---|---|
| Clinical trial registration | |
| Study protocol | |
| Data collection | |
| Outcomes | |

# Dual use research of concern

Policy information about dual use research of concern

## Hazards

Could the accidental, deliberate or reckless misuse of agents or technologies generated in the work, or the application of information presented in the manuscript, pose a threat to:

| | No | Yes |
|---|---|---|
| Public health | ☐ | ☐ |
| National security | ☐ | ☐ |
| Crops and/or livestock | ☐ | ☐ |
| Ecosystems | ☐ | ☐ |
| Any other significant area | ☐ | ☐ |

## Experiments of concern

Does the work involve any of these experiments of concern:

| | No | Yes |
|---|---|---|
| Demonstrate how to render a vaccine ineffective | x | ☐ |
| Confer resistance to therapeutically useful antibiotics or antiviral agents | x | ☐ |
| Enhance the virulence of a pathogen or render a nonpathogen virulent | ☐ | ☑ |
| Increase transmissibility of a pathogen | ☐ | ☑ |
| Alter the host range of a pathogen | ☐ | ☑ |
| Enable evasion of diagnostic/detection modalities | x | ☐ |
| Enable the weaponization of a biological agent or toxin | x | ☐ |
| Any other potentially harmful combination of experiments and agents | x | ☐ |

# Plants

| | |
|---|---|
| Seed stocks | |
| Novel plant genotypes | |
| Authentication | |

# ChIP-seq

## Data deposition

☐ Confirm that both raw and final processed data have been deposited in a public database such as GEO.

☐ Confirm that you have deposited or provided access to graph files (e.g. BED files) for the called peaks.

| | |
|---|---|
| Data access links
*May remain private before publication.* | |
| Files in database submission | |
| Genome browser session
(e.g. UCSC) | |

## Methodology

| | |
|---|---|
| Replicates | |
| Sequencing depth | |
| Antibodies | |
| Peak calling parameters | |
| Data quality | |

Software

## Flow Cytometry

### Plots

Confirm that:

☐ The axis labels state the marker and fluorochrome used (e.g. CD4-FITC).

☐ The axis scales are clearly visible. Include numbers along axes only for bottom left plot of group (a 'group' is an analysis of identical markers).

☐ All plots are contour plots with outliers or pseudocolor plots.

☐ A numerical value for number of cells or percentage (with statistics) is provided.

### Methodology

Sample preparation

Instrument

Software

Cell population abundance

Gating strategy

☐ Tick this box to confirm that a figure exemplifying the gating strategy is provided in the Supplementary Information.

## Magnetic resonance imaging

### Experimental design

Design type

Design specifications

Behavioral performance measures

Imaging type(s)

Field strength

Sequence & imaging parameters

Area of acquisition

Diffusion MRI          ☐ Used          ☐ Not used

### Preprocessing

Preprocessing software

Normalization

Normalization template

Noise and artifact removal

Volume censoring

### Statistical modeling & inference

Model type and settings

Effect(s) tested

Specify type of analysis: ☐ Whole brain ☐ ROI-based ☐ Both

Statistic type for inference

(See Eklund et al. 2016)

Correction

## Models & analysis

| n/a | Involved in the study |
|-----|------------------------|
| ☐ | ☑ Functional and/or effective connectivity |
| ☐ | ☑ Graph analysis |
| ☐ | ☑ Multivariate modeling or predictive analysis |

| | |
|---|---|
| Functional and/or effective connectivity | Sequence alignment analysis |
| Graph analysis | Bar graph analysis on outbreak numbers |
| Multivariate modeling and predictive analysis | Bayesian phylodynamics |

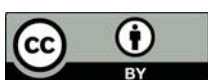

