## [Peer Review File · Nature]

Feline infectious peritonitis epizootic caused by a recombinant coronavirus

Corresponding Author: Dr Christine Tait-Burkard

Version 0:

Reviewer comments:

Referee #1

(Remarks to the Author)

The Authors describe the outbreak/epidemic of feline infectious peritonitis (FIP) coronavirus in terms of epidemiological and clinical patterns in Cyprus, 2023. Using overlapping primer sets, the nearly complete genome of an isolate, FCoV-23, was determined. The virus showed a recombinant nature, with a feline coronavirus backbone and the S gene derived from a pantropic, hypervirulent canine CoV, strain CB05. Sequencing of the S gene of the FIP strains revealed that the Cyprus outbreak was related to the spread of this recombinant virus and that there were several viruses with a truncated S gene, reminiscent of porcine respiratory coronavirus (PRCV).

The study is relevant and describes the emergence of a novel feline coronavirus. The introduction provides a good background. The methodologies are appropriate, the results have been reported clearly, the discussion is balanced and the conclusions are correct. The abstract is comprehensive and clear.

Suggestion:

Line 182: "a single gap of 1,221bp remains in this assembly at the second recombination breakpoint". Have the Authors tried to generate this sequence in the white? They could use several approaches, including SISPA. If they have generated these sequence data, please update the information.

I also have minor comments/suggestions:

Lines 56-57: the grammar should be checked

Line 312-313: the claim "stop this virus epizootic from becoming a panzootic" is clear after reading again and again but is a bit strong for me. I would suggest re-phrasing and toning down.

Referee #2

(Remarks to the Author)

The manuscript reports the emergence of a novel, highly pathogenic and highly transmissible FCoV-CCoV recombinant (FCoV-23) – the cause of the massive feline infectious peritonitis (FIP) outbreak that originated in Cyprus. The findings show that the recombinant bears 97% sequence identity to the previously characterized pantropic canine coronavirus CB/05. Of interest, in the ongoing outbreak FIP development does not seem to be reliant on biotype switch, previously reported to sporadically originate in some FECoV cases following not fully defined de novo (intra-host) mutations. High sequence identity of the FCoV-23 variants from cats in different districts of the island is suggestive of direct transmission. One deletion and several amino acid substitutions in the spike/receptor binding domain may be responsible for the cell tropism shift (high pathogenicity and increased neurotropic potential of FCoV-23). Considering the recent emergence of highly virulent CCoV variants in dogs and CCoV detection in human cases, this research is of high importance. The fact that there is evidence of

the virus spread to the UK further emphasizes the high significance of the current data and investigation. The study is technically sound; however, it is concerning that the final genome assembly/recombination analysis were done based on a consensus sequence (from different samples) rather than on a single representative sample/genome. This needs to be addressed by recovering complete genome sequences from several samples. This should not be a problem considering the very high virus loads reported by the authors.

Another issue is that the manuscript lacks clarity in multiple aspects. For example, the authors state that there is a high sequence identity of FCoV-23 variants detected in different animals, but do not report how high it is. Needs to be corrected. Also, the conclusion based on this observation (e.g. direct transmission) needs to be toned down, because high sequence identity alone is not evidence of direct transmission. Example: CCoV-HuPn-2018 and hCCoV-Z19 sharing 99.4% similarity that were isolated from human cases that took place in different years (2018 and 2017) in Malaysia and Haiti without any evidence of direct transmission.

Please clarify in the text/figure legends the length of the Orf1b, spike, and Orf3c/E/M regions used for the phylogenetic analysis (Figure 2).

The manuscript writing needs some improvement. Please see some concerns below.

Please use abbreviations consistently. Examples of inconsistencies: L153 – “POL1ab and ORF3c/E/M” and L162 – “Orf1b, spike, and Orf3c/E/M”

LL42-43: “High sequence 43 identity of isolates from cats in different districts of the island is strongly supportive of direct” Please replace ‘isolates’ with ‘variants’ or ‘strains’. If the virus was not isolated from a biological sample is not considered to be an isolate.

LL56-58: “The virus exists in two biotypes with the main biotype, feline 57 enteric coronavirus (FECV), showing low virulence and clinical signs are typically limited to mild 58 enteritis.” → “The virus exists in two biotypes with the main biotype, feline 57 enteric coronavirus (FECV), showing low virulence, and clinical signs are typically limited to mild 58 enteritis.”

L61: “...include an abdomen swollen due to peritoneal fluid, fever, weight loss lethargy, anorexia...” → “...include an abdomen swollen due to peritoneal fluid, fever, weight loss, lethargy, anorexia...”

LL97-101: Same sentence repeated twice - “In order for veterinarians 98 to have access to this medication, amongst others, a PCR confirmation was required, reflecting 99 the increased cases seen during August 2023. In order for veterinarians to have access to this medication a PCR confirmation was required, reflecting the increased cases seen during August 101 2023 (Figure 1A & B)”

LL218-219: “Whilst a new mutation each was identified in Orf3a and b, no 219 specific indications of pathogenesis could be determined (Supplementary Table 21).” Please omit ‘each’, not needed here.

LL255-258: These sentences are redundant, please rephrase - “The risk of spreading this outbreak is significant as evidenced by the first confirmed UK case40 256 . This is exemplified by the 257 recent confirmation of a first UK-imported case with further investigations into other cases 258 ongoing”

LL259: “CCoV infections in dogs are typically self-limiting, producing mild or asymptomatic enteritis.” Enteritis cannot be asymptomatic. If there is enteritis, the infection is not asymptomatic. Please rephrase.

Referee #3

(Remarks to the Author)

This manuscript reports a highly pathogenic FCoV-CCoV recombinant responsible for a rapidly spreading outbreak of feline infectious peritonitis (FIP). The importance of the topic is undoubtedly clear – while recombination events in coronaviruses are not surprising, as the Authors acknowledge, the fact that this resulted in a virus with enhanced transmissibility of FIP requires attention and in-depth characterization. The Authors make a good effort to bring this to the attention to the field with the data at hand, but the characterization remains limited. I would not expect the Authors to be able to gain clear insights into shedding routes and virulence factors for example, but a more comprehensive genomic characterization seems in order. The focus has been on the spike gene and no single genome is currently available; only a population genome for which a part is still missing (a part that is important to delineate the recombination event and to characterize key determinant mutations of FIPV). The Authors are aware of the limitations as reflected in the cautionary language and statements that allude to further research. The data availability statement also refers to ongoing work and hopefully more data to be produced. So, I consider the current report preliminary and really hope the Authors can provide more substantial data. While I appreciate for example sample preservation until a working primer scheme is developed for amplicon sequencing, there are also metagenomic protocols that could be used to produce genomes for at least a few samples. A better genomic characterization could offer ways to reconstruct the dynamics of spread, and protocols to achieve this, would allow others to build on the work; it would facilitate genomic epidemiology to track the spread of the virus and hence make the work of the Authors far more impactful.

Minor comments

Both the abstract and the discussion refer to high sequence identity of isolates from cats as compelling evidence supportive of direct transmission, but I was unable to find where this evidence is presented.

The taxa labels in the trees in Figure 3 are not legible. I would recommend to remove the, refer to the SI for bigger trees with labels and color actual clades/branches or tip symbols instead in the trees in the main figure. In doing so, it would be useful to also roughly use the same orientation for the FCoV and CCoV clusters.

As recombination represents a major violation for phylogenetic tree reconstruction, I find it inappropriate to (first) refer to a phylogeny that ignores this (Supplementary Figure 5) and then show the recombination analysis in Fig. 3. I would recommend to focus on the recombination analysis and only show trees for non-recombinant subregions if needed.

It would be useful to use the same colors to represent pCCoV, FCoV and FCoV in Figure 2 and 3.

line 40. "The recombination, spanning spike, shows 97% sequence identity to the pantropic canine coronavirus CB/05." I consider recombination to be an event or a mosaic genome as a result of the event, so it is unclear why part of the mosaic genome should be called 'the recombinant'. I would recommend to refer to the 'minor recombinant region' in this case.

lines 97-99 & 99-101: repetition

Referee #4

(Remarks to the Author)

I was specifically asked to comment on the genomic analyses, which I will take in a broad sense here to cover the process of sequence generation and the analyses until the production of a genome and/or shorter sequences.

The authors have mentioned themselves that they are still generating data to better support their claims, and this review was therefore written to provide them with external feedback/advice while they are at it.

First a summary of what the authors did:

After confirming infection with a qPCR, the authors ran a multiplex PCR system designed for whole genome amplification of FCoV. The resulting amplicons were sequenced on an ONT platform. This did not allow for the recovery of a complete genome, but the authors aggregated all reads generated across individuals to reconstruct a large fraction thereof ("at a population level" [sic]). The provisional draft genome is available as Sup Data, and the authors used it to identify a recombinant fragment spanning the end of orf1b and a big part of spike.

The authors were in the position to use some of the well-functioning PCR systems to recover partial sequences in three regions of the genome from several dozens of specimens (also based on ONT sequencing of the amplicons): i) in orf1b upstream of the first putative recombination breakpoint, ii) end of orf1b+first half of spike downstream of the first putative recombination breakpoint and up stream of the (yet not positioned) second recombination breakpoint, iii) end of 3c+E+half of M downstream of the second breakpoint. The corresponding three sequence files are provided by the authors as Sup Data.

My assessment:

At the moment the genomic evidence presented is relatively weak/insufficient.

What the authors want to be in the position to do here is to exclude the alternative scenario, whereby what they detected is coinfection with FCoV and CCoV. It is not a totally implausible scenario –coinfection with enteric viruses is frequent and it is an aggravating factor for disease severity and outcomes (for example when it comes to coinfection with CCoV and canine parvovirus in dogs).

Unfortunately, the way they tried to generate full genomes from positive samples really does not help because it is amplicon-based and therefore opens the door to different PCR systems having different biases and preferentially amplifying FCoV or CCoV. From a quick look at primer sequences it does not seem very likely but it still worrying enough that I would very much favor not using a amplicon-based strategy, and rather switch to RNAseq.

RNAseq is very likely to work well since histopathology suggests high viral loads (which their qPCR maybe confirm for some specimens?) and it would of course be much less biased than an amplicon sequencing strategy. It would also allow for a broader assessment of the potential involvement of other enteric agents (which is currently not really discussed by the authors – there is no attempt at differential diagnosis reported in the manuscript or the recently published paper, as far as I can tell). An additional argument for switching technique is the very poor performance of amplicon sequencing in this case. Pooling all their sequencing attempts the authors cannot even close the genome: their sequence still comprises 6.7% ambiguous bases (including in a crucial region of spike), and a significant fraction of the remaining 93.3% unambiguous bases is probably supported by very few reads (as per the authors' own account).

Irrespective of how they do it, the authors clearly need to produce a clean genome from a single infected individual to definitely prove their claim. This should be based on a high quality map, where the recombination breakpoints are well covered by numerous unique reads and no patterns suggestive of a technical artefact can be observed. The current Sup Fig 7 is really unconvincing: it only places a single read in the region supposed to cover one of the breakpoints. It is not enough because: i) a sequence alignment should be produced where bases are legible, ii) a single read recombinant read might as well be an isolated jumping PCR artefact.

In a nutshell for this core part: i) the authors should produce at least one high quality genome from a single case, ii) they would probably be much better off using RNAseq to do so, which would considerably reinforce their overall claim, both with respect to validating the recombinant nature of this virus and reinforcing its link to this disease wave.

I would like to add observations regarding the spike alignment/deletion that I also quickly checked: i) the sequences reported are incomplete, only starting 100nt into spike (when I map the forward primer of this system it lands a good 600bp upstream), ii) the authors refer to a deletion in spike, and reading them I expected that this would represent a single event (so same start and end position; the authors mention "two versions of this spike gene, one of which has a deletion of approximately 630 nt") – it is not at all the case and nearly every other sequence presents with a different deletion in the same zone (by a rough estimate about 20 different deletions of 456-723 nt in this region). This clearly warrants further confirmation/investigation (good-quality genomes should help a lot here).

Best regards,

Sebastien Calvignac-Spencer

Version 1:

Reviewer comments:

Referee #1

(Remarks to the Author)

The Authors have fixed the major problem of the study, i.e. they have been able to generate the complete genome sequence of more feline coronavirus strains.

Also, they have fixed/resolved minor issues scattered in the manuscript, as noted by me and other referees.

Referee #2

(Remarks to the Author)

The newly generated data including the complete genome and partial sequences, the associated analyses as well as all other revisions have greatly improved the manuscript quality. The authors did well addressing the original criticism. I find it suitable for publication

Referee #3

(Remarks to the Author)

The Authors have largely addressed my concerns. In particular, the new full genome data increases the impact of the study significantly. A few remaining minor comments:

The addition of the higher resolution figures is useful given that there is still a problem with the legibility of the taxa labels in main Figure 2. I do not understand the insistence to include labels that cannot be properly read in the main figure.

In reply to my comment on phylogenetic inference of recombinant data, the Authors indicate that the trees focus on non-recombinant subregions. However, unless I am missing something, the tree that I was referring to (Supplementary Fig. 5) is based on complete genome data (as mentioned in its caption) and thus completely ignores recombination which invalidates the application of a strictly bifurcating tree.

Referee #4

(Remarks to the Author)

In this revision, the authors present significantly more data than in the first version of the paper and provide a slightly more detailed analysis of the deletion pattern in the S of FCoV-23. The text itself has also improved.

However, I feel that there are major shortcomings and limitations that should be addressed before publishing this work.

Major concerns:

1. Potential syndemic scenario:

My primary concern remains the possibility that the signal detected by the authors represents not a recombinant virus, but rather a syndemic—co-circulation of two viruses in cats—and that they might have inadvertently assembled distinct genomes into a single genome due to the sequencing strategy employed. The authors' response does not really address this concern. To be clear: the problem is not amplicon sequencing per se – I also use this strategy a lot where it makes sense. However, amplicon-based sequencing always comes with the risk of unintended amplification biases and if the authors' preliminary data is generated with amplicons then there is the potential for an initial bias that further fine-tuning of secondary systems can by definition only reproduce.

What the authors should do:

- Provide the sequences of the exact amplicons that cover the recombination breakpoints (e.g., pair 36A and 43). These contiguous sequences should display FCoV-like characteristics on one side and CCoV-like on the other, or vice versa.
- If the multiplex amplicon system aligns with the boundaries of the recombinant region, present new data showing reads that span the recombination breakpoints, perhaps by elaborating on the system that the authors say they used for the long-range PCRs spanning orf1b and S.
- Add a section or a few sentences in the article that explicitly consider the possibility of a syndemic.

2. Analysis of deletions in S and phylogenomic analyses:

The investigation of deletions in the S protein and the phylogenomic analyses could be enhanced to provide deeper insights.

What the authors should do:

- Map these deletions onto full-genome phylogenetic trees (excluding the recombinant fragment) to identify potential clusters indicative of transmission. Potentially, perform formal ancestral character state reconstruction to trace the evolution of the deletions.
- Compute the average patristic distance of genomes harboring the same deletion versus those without. If transmission of viruses with specific deletions is occurring, genomes with the same deletion should have a lower average distance.
- Implement more sophisticated phylodynamic analyses, such as estimating the date of the recombination event using molecular clock models.

Minor concerns/specific suggestions:

- Line 34: Consider deleting "and demonstrated" for conciseness.
- Lines 73-74: Clarify what is meant by "gradual evolution."
- Line 77: Correct the statement to reflect that FIP is not transmissible, but FCoVs are.
- Line 79: Capitalize and italicize the genus name for proper taxonomic formatting.
- Line 99: Replace "reflecting" with "which probably explains" or a similar phrase for clarity.
- Lines 106-108: The distribution could also be interpreted as a long tail to the first wave and/or endemic circulation in an exposed population. Consider toning down the "three waves" narrative, especially given the fluctuating testing reported. Did the age distribution change between the peak in mid-2023 and 2024? Including this information could be insightful.
- Lines 139-140: The phrase "with Illumina sequencing" is misleading. What you really mean is "shotgun sequencing". Clarify that estimates of FCoV versus background RNA concentrations prompted the use of an enrichment strategy, independent of the sequencing platform (the authors could have sequenced your amplicons on an Illumina instrument, the sequencing platform is essentially irrelevant).
- Line 145: Provide basic statistics comparing the 19 full-length genomes before delving into more elaborate phylogenomic analyses (as suggested above).
- Lines 163-165: The last sentence can be removed as readers are typically familiar with scales.
- Figure 2: The labels in the trees are difficult to read and could be replaced with simple colored squares or symbols. Enlarging the trees or providing zoomed-in sections, particularly of the areas representing FCoV-23, would enhance readability.
- Lines 196-199: This analysis may not add substantial value as it reports averages of separate evolutionary trajectories. Consider focusing on more informative phylodynamic analyses as suggested earlier.
- Lines 272-273: If no other cases abroad have been reported thus far, mention this to suggest that the actual risk may be low.
- Line 304: Remove "earlier discussed".
- Lines 308-319: This paragraph repeats earlier arguments. Consider integrating the few new points into previous sections to avoid redundancy, and delete it here entirely.

Sebastien Calvignac-Spencer

Version 2:

Reviewer comments:

Referee #4

(Remarks to the Author)

The authors have provided the additional information about the recombination breakpoints that I considered critical, which is good.

[In passing, the text bit about a potential syndemic and the associated justification is weird. A syndemic might not be a very plausible explanation, agreed, but clearly there is no reason to make such an event conditional on human interventions - nature does surprising things too. I am totally fine if the authors prefer not to mention a syndemic as an alternative scenario, but then they should explain more simply that "[They] excluded the possibility that this apparent recombination event would only reflect coinfection with two viruses and preferential amplification effects in [their] multiplex PCR assay by examining the relative positions of recombination breakpoints and PCR primers. Amplicons 32 and in the tiled amplification scheme span

the recombination breakpoints and show clear FCoV|pCCoV and pCCoV|FCoV characteristics in the individual amplicons, respectively, which validates the recombination event."]

In addition, the authors have also added, or tried to add, more analyses. All in all, the paper has improved, and I think it is now technically sound .

The text is still quite rough around the edges though, with typos (eg non capitalization of the virus species name in the last paragraph - btw please update virus species name; the ICTV has now moved to Latin binomials, incl. for CoVs), unusual words (eg reposit) and awkward phrasing. I would recommend that it be carefully edited, and my feeling is that it could also be condensed significantly.

Best regards,

Sebastien Calvignac-Spencer

Reply to the reviewer's comments:

We appreciate the reviewers' comments to our manuscript and have revised our manuscript as follows:

Referee #1 (Remarks to the Author):

The Authors describe the outbreak/epidemic of feline infectious peritonitis (FIP) coronavirus in terms of epidemiological and clinical patterns in Cyprus, 2023. Using overlapping primer sets, the nearly complete genome of an isolate, FCoV-23, was determined. The virus showed a recombinant nature, with a feline coronavirus backbone and the S gene derived from a pantropic, hypervirulent canine CoV, strain CB05. Sequencing of the S gene of the FIP strains revealed that the Cyprus outbreak was related to the spread of this recombinant virus and that there were several viruses with a truncated S gene, reminiscent of porcine respiratory coronavirus (PRCV).

The study is relevant and describes the emergence of a novel feline coronavirus. The introduction provides a good background. The methodologies are appropriate, the results have been reported clearly, the discussion is balanced and the conclusions are correct. The abstract is comprehensive and clear.

We thank the reviewer for this complementing feedback.

Suggestion:

Line 182: "a single gap of 1,221bp remains in this assembly at the second recombination breakpoint". Have the Authors tried to generate this sequence in the while? They could use several approaches, including SISPA. If they have generated these sequence data, please update the information.

Significantly increased data has been provided in the updated manuscript, including 20 full genome sequences, 10 of which without any single gap or uncertain region. Further partial sequences have been generated to assess the validity of our hypotheses.

I also have minor comments/suggestions:

Lines 56-57: the grammar should be checked

Done

Line 312-313: the claim "stop this virus epizootic from becoming a panzootic" is clear after reading again and again but is a bit strong for me. I would suggest re-phrasing and toning down.

This sentence has now been changed to: "Prevention of spread and the development of new vaccines are important to stop this epizootic virus from spreading beyond Cyprus."

Referee #2 (Remarks to the Author):

The manuscript reports the emergence of a novel, highly pathogenic and highly transmissible FCoV-CCoV recombinant (FCoV-23) – the cause of the massive feline infectious peritonitis (FIP) outbreak that originated in Cyprus. The findings show that the recombinant bears 97% sequence identity to the previously characterized pantropic canine coronavirus CB/05. Of interest, in the ongoing outbreak FIP development does not seem to be reliant on biotype switch, previously reported to sporadically originate in some FCoV cases following not fully defined de novo (intra-host) mutations. High sequence identity of the FCoV-23 variants from cats in different districts of the island is suggestive of direct transmission. One deletion and several amino acid substitutions in the spike/receptor binding domain may be responsible for the cell tropism shift (high pathogenicity and increased neurotropic potential of FCoV-23). Considering the recent emergence of highly virulent CCoV variants in dogs and CCoV detection in human cases, this research is of high importance. The fact that there is evidence of the virus spread to the UK further emphasizes the high significance of the current data and investigation. The study is technically sound; however, it is concerning that the final genome assembly/recombination analysis were done based on a consensus sequence (from different samples) rather than on a single representative sample/genome. This needs to be addressed by recovering complete genome sequences from several samples. This should not be a problem considering the very high virus loads reported by the authors.

*Another issue is that the manuscript lacks clarity in multiple aspects. For example, the authors state that there is a **high sequence identity of FCoV-23 variants detected in different animals, but do not report how high it is**. Needs to be corrected.*

Further information towards this has been provided in supplementary figures S9 and S10.

*Also, the conclusion based on this observation (e.g. **direct transmission**) needs to be toned down, because high sequence identity alone is not evidence of direct transmission. Example: CCoV-HuPn-2018 and hCCoV-Z19 sharing 99.4% similarity that were isolated from human cases that took place in different years (2018 and 2017) in Malaysia and Haiti without any evidence of direct transmission.*

This has been caveated to state:

“Our data suggest that there is direct transmission of FCoV-23 between cats based on high sequence identity (>99.17% genome-scale including varying S-deletion length, Supplementary figures S9 & S10), high viral loads (RT-qPCR only) in feces, as well as the wave-like movement of disease across the island.”

Please clarify in the text/figure legends the length of the Orf1b, spike, and Orf3c/E/M regions used for the phylogenetic analysis (Figure 2).

This has been specified in lines 145 fff:

“This yielded a total of 19 full-length genomes, 45 partial sequences spanning a section of orf1b (~1,000 bp), 63 spanning the first part of S (~2,250 bp), and 42 spanning ORFs 3c/E(4)/M(5) (~1,000 bp).”
Genome and grey bars in figure 2 have furthermore been updated to be to scale.

The manuscript writing needs some improvement. Please see some concerns below.

Please use abbreviations consistently. Examples of inconsistencies: L153 – “POL1ab and ORF3c/E/M” and L162 – “Orf1b, spike, and Orf3c/E/M”

Corrected

LL42-43: “High sequence 43 identity of isolates from cats in different districts of the island is strongly supportive of direct” Please replace ‘isolates’ with ‘variants’ or ‘strains’. If the virus was not isolated from a biological sample is not considered to be an isolate.

This has been changed to samples.

LL56-58: “The virus exists in two biotypes with the main biotype, feline 57 enteric coronavirus (FECV), showing low virulence and clinical signs are typically limited to mild 58 enteritis.” → “The virus exists in two biotypes with the main biotype, feline 57 enteric coronavirus (FECV), showing low virulence, and clinical signs are typically limited to mild 58 enteritis.”

Changed

L61: “...include an abdomen swollen due to peritoneal fluid, fever, weight loss lethargy, anorexia...” → “...include an abdomen swollen due to peritoneal fluid, fever, weight loss, lethargy, anorexia...”

Changed

LL97-101: Same sentence repeated twice - “In order for veterinarians 98 to have access to this medication, amongst others, a PCR confirmation was required, reflecting 99 the increased cases seen during August 2023. In order for veterinarians to have access to this medication a PCR confirmation was required, reflecting the increased cases seen during August 101 2023 (Figure 1A & B)”

Removed

LL218-219: “Whilst a new mutation each was identified in Orf3a and b, no 219 specific indications of pathogenesis could be determined (Supplementary Table 21).” Please omit ‘each’, not needed here.

Removed

LL255-258: These sentences are redundant, please rephrase - “The risk of spreading this outbreak is significant as evidenced by the first confirmed UK case 256 . This is exemplified by the 257 recent confirmation of a first UK-imported case with further investigations into other cases 258 ongoing”

Sorted

LL259: “CCoV infections in dogs are typically self-limiting, producing mild or asymptomatic enteritis.” Enteritis cannot be asymptomatic. If there is enteritis, the infection is not asymptomatic. Please rephrase.

Rephrased: CCoV infections in dogs are typically self-limiting, producing mild enteritis or presenting asymptotically.

Referee #3 (Remarks to the Author):

*This manuscript reports a highly pathogenic FCoV-CCoV recombinant responsible for a rapidly spreading outbreak of feline infectious peritonitis (FIP). The importance of the topic is undoubtedly clear – while recombination events in coronaviruses are not surprising, as the Authors acknowledge, the fact that this resulted in a virus with enhanced transmissibility of FIP requires attention and in-depth characterization. The Authors make a good effort to bring this to the attention to the field with the data at hand, but the characterization remains limited. I would not expect the Authors to be able to gain clear insights into shedding routes and virulence factors for example, but a **more comprehensive genomic characterization** seems in order. The focus has been on the spike gene and no single genome is currently available; only a population genome for which a part is still missing (a part that is important to delineate the recombination event and to characterize key determinant mutations of FIPV). The Authors are aware of the limitations as reflected in the cautionary language and statements that allude to further research. The data availability statement also refers to ongoing work and hopefully more data to be produced. **So, I consider the current report preliminary and really hope the Authors can provide more substantial data.** While I appreciate for example sample preservation until a working primer scheme is developed for amplicon sequencing, there are also metagenomic protocols that could be used to produce genomes for at least a few samples. **A better genomic characterization could offer ways to reconstruct the dynamics of spread, and protocols to achieve this, would allow others to build on the work; it would facilitate genomic epidemiology to track the spread of the virus and hence make the work of the Authors far more impactful.***

Significantly increased data has been provided in the updated manuscript, including 20 full genome sequences, 10 of which without any single gap or uncertain region. Further partial sequences have been generated to assess the validity of our hypotheses.

Minor comments

Both the abstract and the discussion refer to high sequence identity of isolates from cats as compelling evidence supportive of direct transmission, but I was unable to find where this evidence is presented.

Further information towards this has been provided in supplementary figures S9 and S10.

The taxa labels in the trees in Figure 3 are not legible. I would recommend to remove the, refer to the SI for bigger trees with labels and color actual clades/branches or tip symbols instead in the trees in the main figure. In doing so, it would be useful to also roughly use the same orientation for the FCoV1 and CCoV1 clusters.

We have tried to improve both the resolution in the figure itself as well as the colors used for presentation. We have still added the higher resolution trees to improve contrast for improved legibility if people would like to investigate the tree further.

As recombination represents a major violation for phylogenetic tree reconstruction, I find it inappropriate to (first) refer to a phylogeny that ignores this (Supplementary Figure 5) and then show the recombination analysis in Fig. 3. I would recommend to focus on the recombination analysis and only show trees for non-recombinant subregions if needed.

The trees are in fact focussing on non-recombinant subregions, highlighting exactly that non-recombinant subregions in orf1b and spanning orf3c/E(4)/M(5) are closely related to FCoV-1, and that the S subregion is closest related to CCoVs. The recombination analysis across the whole genome then confirms that recombination occurs spanning the end of orf1b and S.

It would be useful to use the same colors to represent pCCoV, FCoV1 and FCoV1 in Figure 2 and 3.

For the sake of best legibility we have not used the same colors for Figures 2 and 3. Whilst the pastel colours in figure 2 permit best legibility of labels, they don't work well to discern the different lines in figure 3. We have therefore opted for significantly contrasting colors in figure 3.

line 40. “The recombination, spanning spike, shows 97% sequence identity to the pantropic canine coronavirus CB/05.” I consider recombination to be an event or a mosaic genome as a result of the event, so it is unclear why part of the mosaic genome should be called ‘the recombinant’. I would recommend to refer to the ‘minor recombinant region’ in this case.

This sentence has been updated according to the reviewer’s suggestion and also updated to correctly reference NA/09. CB/05 was equally closely related when spike was incomplete. Now NA/09 is more closely related than the full S sequence has been established: “The minor recombinant region, spanning spike (S), shows 96.5% sequence identity to the pantropic canine coronavirus NA/09.”

lines 97-99 & 99-101: repetition

Removed

Referee #4 (Remarks to the Author):

I was specifically asked to comment on the genomic analyses, which I will take in a broad sense here to cover the process of sequence generation and the analyses until the production of a genome and/or shorter sequences.

The authors have mentioned themselves that they are still generating data to better support their claims, and this review was therefore written to provide them with external feedback/advice while they are at it.

First a summary of what the authors did:

After confirming infection with a qPCR, the authors ran a multiplex PCR system designed for whole genome amplification of FCoV. The resulting amplicons were sequenced on an ONT platform. This did not allow for the recovery of a complete genome, but the authors aggregated all reads generated across individuals to reconstruct a large fraction thereof (“at a population level” [sic]). The provisional draft genome is available as Sup Data, and the authors used it to identify a recombinant fragment spanning the end of orf1b and a big part of spike.

The authors were in the position to use some of the well-functioning PCR systems to recover partial sequences in three regions of the genome from several dozens of specimens (also based on ONT sequencing of the amplicons): i) in orf1b upstream of the first putative recombination breakpoint, ii) end of orf1b+first half of spike downstream of the first putative recombination breakpoint and up stream of the (yet not positioned) second recombination breakpoint, iii) end of 3c+E+half of M downstream of the second breakpoint. The corresponding three sequence files are provided by the authors as Sup Data.

My assessment:

At the moment the genomic evidence presented is relatively weak/insufficient.

What the authors want to be in the position to do here is to exclude the alternative scenario, whereby what they detected is coinfection with FCoV and CCoV. It is not a totally implausible scenario –coinfection with enteric viruses is frequent and it is an aggravating factor for disease severity and outcomes (for example when it comes to coinfection with CCoV and canine parvovirus in dogs).

Unfortunately, the way they tried to generate full genomes from positive samples really does not help because it is amplicon-based and therefore opens the door to different PCR systems having different biases and preferentially amplifying FCoV or CCoV. From a quick look at primer sequences it does not seem very likely but it still worrying enough that I would very much favor not using a amplicon-based strategy, and rather switch to RNAseq.

Amplicon-based sequencing methods are very common place these days (e.g. large-scale sequencing of SARS-CoV-2 or Ebola) and we disagree with the reviewer that they are inferior to for example paired-end Illumina-based sequencing, which is equally assembly-based and allows for exactly the same errors, i.e. the assembly of a potential co-infection of FCoV and CCoV to be detected. Also, with Illumina sequencing methods, it is known for FIPV that read depth from samples is usually insufficient. Therefore, some groups rely on vial genome-based capture methods (<https://doi.org/10.1101/2024.01.02.573944>), which are only successful or an option if one knows the target.

What we have done here is not a priori amplicon-based. We have generated an initial consensus based on amplification as well as cDNA basis. However, cDNA reading depth from the RNA extracts provided was far too low to generate a reliable consensus. Therefore, we worked with a combination of very long amplicons alongside the initial 1kb amplicons, some of which are 6kb long. This way, we could generate a confident consensus, that clearly shows the FCoV/CCoV recombination independent of a co-infection in the same molecule.

RNAseq is very likely to work well since histopathology suggests high viral loads (which their qPCR maybe confirm for some specimens?) and it would of course be much less biased than an amplicon sequencing strategy. It would also allow for a broader assessment of the potential involvement of other enteric agents (which is currently not really discussed by the authors – there is no attempt at differential diagnosis reported in the manuscript or the recently published paper, as far as I can tell). An additional argument for switching technique is the very poor performance of amplicon sequencing in this case. Pooling all their sequencing attempts the authors cannot even

close the genome: their sequence still comprises 6.7% ambiguous bases (including in a crucial region of spike), and a significant fraction of the remaining 93.3% unambiguous bases is probably supported by very few reads (as per the authors' own account).

The samples we have accessible from cats are very small, surplus diagnostic samples. From most cats, we only have access to 30-40 µl of extracted NA from a clinical sample. Whilst the viral load at the beginning may be high, what can be extracted from the samples is very small amounts of virus. This is a common problem for the sequencing of FIPV (as discussed in <https://doi.org/10.1101/2024.01.02.573944>) without prior amplification of the virus (which defeats the purpose of sequencing field/cat samples).

Whilst we agree that 6.7% ambiguous bases is high, we would also like to highlight that ambiguous bases are biologically relevant in viruses and often correct in the context of a virus quasispecies due to errors in RNA copying.

Irrespective of how they do it, the authors clearly need to produce a clean genome from a single infected individual to definitely prove their claim. This should be based on a high quality map, where the recombination breakpoints are well covered by numerous unique reads and no patterns suggestive of a technical artefact can be observed. The current Sup Fig 7 is really unconvincing: it only places a single read in the region supposed to cover one of the breakpoints. It is not enough because: i) a sequence alignment should be produced where bases are legible, ii) a single read recombinant read might as well be an isolated jumping PCR artefact.

We hope with the additional information on how sequencing was achieved, plus the additional full-genome sequences and partial sequences provided we could convince the reviewer that this recombination is real. It occurs in the samples from all cats analyzed so far (>63). For most of these, we have also since performed long-range PCRs spanning orf1b and S prior to running amplicon-based sequencing to determine the presence of the domain 0 deletion mutant.

In a nutshell for this core part: i) the authors should produce at least one high quality genome from a single case, ii) they would probably be much better off using RNAseq to do so, which would considerably reinforce their overall claim, both with respect to validating the recombinant nature of this virus and reinforcing its link to this disease wave.

As highlighted above, RNAseq (for example using Illumina-based sequencing) for these samples is not an option due to the low amount of sample available to us and also based on previous experience sequencing FIPV. This is further highlighted by the struggles faced by other labs who now use hybridization capture for enrichment. This is not feasible with such low amounts of sample and/or not knowing the target. As furthermore highlighted, we have tried this using direct cDNA sequencing with Nanopore-based methods (to see if it were feasible and worthwhile going to Illumina) and had to discount that strategy. The very limited information from cDNA sequencing was however used to validate the recombinant and optimise the amplicon-based sequencing method.

We can now present 10 gap free genomes and 10 genomes with small "N" regions alongside 150 partial sequences used for phylogenetic analysis.

I would like to add observations regarding the spike alignment/deletion that I also quickly checked: i) the sequences reported are incomplete, only starting 100nt into spike (when I map the forward primer of this system it lands a good 600bp upstream), ii) the authors refer to a deletion in spike, and reading them I expected that this would represent a single event (so same start and end position; the authors mention "two versions of this spike gene, one of which has a deletion of approximately 630 nt") – it is not at all the case and nearly every other sequence presents with a different deletion in the same zone (by a rough estimate about 20 different deletions of 456-723 nt in this region). This clearly warrants further confirmation/investigation (good-quality genomes should help a lot here).

The domain 0 deletion was indeed a conundrum that we couldn't figure out at first. Only when we had good quality reads (>50 validated by Sanger sequencing) we can now confirm that this is a likely in-host deletion that is highly variable.

Reply to the reviewer's comments:

We appreciate the reviewers' comments to our manuscript and have revised our manuscript as follows:

Referee #1 (Remarks to the Author):

The Authors have fixed the major problem of the study, i.e. they have been able to generate the complete genome sequence of more feline coronavirus strains.

Also, they have fixed/resolved minor issues scattered in the manuscript, as noted by me and other referees.

Referee #2 (Remarks to the Author):

The newly generated data including the complete genome and partial sequences, the associated analyses as well as all other revisions have greatly improved the manuscript quality. The authors did well addressing the original criticism. I find it suitable for publication

Referee #3 (Remarks to the Author):

The Authors have largely addressed my concerns. In particular, the new full genome data increases the impact of the study significantly. A few remaining minor comments:

The addition of the higher resolution figures is useful given that there is still a problem with the legibility of the taxa labels in main Figure 2. I do not understand the insistence to include labels that cannot be properly read in the main figure.

We have updated this figure to make it more legible – focussing on the GB number only as a label, highlighting other properties in a coloured ring on the outside. Consequently, we have now removed supplementary figures previously identified as Supplementary figures S2-S5.

Furthermore, we have removed some sequences used for the alignments in figure 2 based on in-depth metadata analysis; any laboratory isolate strains, vaccines, or infection experiment sequences were removed.

In reply to my comment on phylogenetic inference of recombinant data, the Authors indicate that the trees focus on non-recombinant subregions. However, unless I am missing something, the tree that I was referring to (Supplementary Fig. 5) is based on complete genome data (as mentioned in its caption) and thus complete ignores recombination which invalidates the application of a strictly bifurcating tree.

This is correct. We have used the tree to highlight that the full genome of FCoV-23 still falls largely close to FCoV1 with the exception of the spike region.

Referee #4 (Remarks to the Author):

In this revision, the authors present significantly more data than in the first version of the paper and provide a slightly more detailed analysis of the deletion pattern in the S of FCoV-23. The text itself has also improved.

However, I feel that there are major shortcomings and limitations that should be addressed before publishing this work.

Major concerns:

1. Potential syndemic scenario:

My primary concern remains the possibility that the signal detected by the authors represents not a recombinant virus, but rather a syndemic—co-circulation of two viruses in cats— and that they might have inadvertently assembled distinct genomes into a single genome due to the sequencing strategy employed. The authors' response does not really address this concern. To be clear: the problem is not amplicon sequencing per se – I also use this strategy a lot where it makes sense. However, amplicon-based sequencing always come with the risk of

unintended amplification biases and if the authors' preliminary data is generated with amplicons then there is the potential for an initial bias that further fine-tuning of secondary systems can by definition only reproduce

We thank the reviewer for their constructive feedback and clarification of their suspicions of a syndemic scenario. However, a syndemic on this scale is basically only realistically possible if there were a vaccination effort in the background, which is not occurring for neither FCoV or CCoV. Having two viruses circulate at the same speed in the same setting would be very novel and even more concerning than the events described.

What the authors should do:

- Provide the sequences of the exact amplicons that cover the recombination breakpoints (e.g., pair 36A and 43). These contiguous sequences should display FCoV-like characteristics on one side and CCoV-like on the other, or vice versa.
- If the multiplex amplicon system aligns with the boundaries of the recombinant region, present new data showing reads that span the recombination breakpoints, perhaps by elaborating on the system that the authors say they used for the long-range PCRs spanning orf1b and S.

Figure 1: Site view of the pCCoV recombination in the FCoV-23 genome. Amplicons in the tiled amplicon array spanning the pCCoV recombination boundaries highlighted.

As can be seen above, there are three amplicons in the scheme that all span the recombination sites. Both amplicons 32 and 41 cover large sections of the FCoV backbone and the pCCoV recombination. Being covered by a single amplicon and not by a conjunction making a hybrid syndemic “Frankenstein” virus highly unlikely. We don’t think this information is something that needs added to the manuscript but is a due diligence of methodology. We had already validated this fact immediately after the initial recombination analyses highlighted an unusual spike in a blast search.

- Add a section or a few sentences in the article that explicitly consider the possibility of a syndemic.

The following sentences have been added to the manuscript lines 193ff:

Whilst there may be a negligible chance of a syndemic, i.e. the circulation of two viruses in all of these cats at the same time, this is only a realistic scenario in this case if there were vaccination efforts ongoing, which there aren’t. Furthermore, amplicons 32 and 41 in the tiled amplification scheme span the recombination breakpoints and show clear FCoV|pCCoV and pCCoV|FCoV characteristics in the individual amplicons, respectively.

2. Analysis of deletions in S and phylogenomic analyses:

The investigation of deletions in the S protein and the phylogenomic analyses could be enhanced to provide deeper insights.

What the authors should do:

- Map these deletions onto full-genome phylogenetic trees (excluding the recombinant fragment) to identify potential clusters indicative of transmission. Potentially, perform formal ancestral character state reconstruction to trace the evolution of the deletions.
- Compute the average patristic distance of genomes harboring the same deletion versus those without. If transmission of viruses with specific deletions is occurring, genomes with the same deletion should have a lower average distance.

With the still limited number of full-genome sequences (20) we have instead generated Bayesian time-resolved phylogenetic trees on the 1ab & 3EM sequences from the Cyprus outbreak to address the question around in-cat versus cat-to-cat transmission of the spike deletion. This information has been added as supplementary figure S6 and details in lines 215 ff.

- Implement more sophisticated phylodynamic analyses, such as estimating the date of the recombination event using molecular clock models.

We have attempted to perform more sophisticated phylodynamic analyses of the recombination in spike but the patchiness in sequencing and the significant lack of detailed metadata have made this problematic. Specifically, for feline coronaviruses there are 128 (near) full-length genome sequences deposited, 31 of those are lab strains/isolates or experimental infections, 9 are of very poor quality (many large gaps), 40 of them are all from the Netherlands between 2007-2010, and 14 from metagenomics alignments (Australia), leaving all but 34 slightly more diverse sequences. Therefore, we cannot estimate the recombination event using molecular clock models without major error.

Figure 2: Global distribution of (near) complete FCoV genomes. 12 countries only, 5 European (Germany, Netherlands, United Kingdom, Belgium, Denmark), 5 Asian (Taiwan, Japan, China, Thailand, India), Australia, and the USA.

Figure 3: Temporal distribution of (near) complete FCoV genomes. Clustered around individual studies in the Netherlands (2007-2010) and Australia (2017-2018).

Minor concerns/specific suggestions:

- Line 34: Consider deleting "and demonstrated" for conciseness.

Deleted

- *Lines 73-74: Clarify what is meant by "gradual evolution."*

Changed to "stepwise"

- *Line 77: Correct the statement to reflect that FIP is not transmissible, but FCoV are.*

This is not quite true. Whilst FIP has limited transmissibility it is not completely non-transmissible.

- *Line 79: Capitalize and italicize the genus name for proper taxonomic formatting.*

Updated to the correct name and formatted as correct.

- *Line 99: Replace "reflecting" with "which probably explains" or a similar phrase for clarity.*

Done

- *Lines 106-108: The distribution could also be interpreted as a long tail to the first wave and/or endemic circulation in an exposed population. Consider toning down the "three waves" narrative, especially given the fluctuating testing reported. Did the age distribution change between the peak in mid-2023 and 2024? Including this information could be insightful.*

The age information is provided and linked to in the text and supplementary tables and no, the age distribution did not change as seen from median and mean. Whilst the reporting is limited, the treatment numbers we receive back from veterinarian surveys do show a wave-like pattern, which is why we've not changed the wording here.

- *Lines 139-140: The phrase "with Illumina sequencing" is misleading. What you really mean is "shotgun sequencing". Clarify that estimates of FCoV versus background RNA concentrations prompted the use of an enrichment strategy, independent of the sequencing platform (the authors could have sequenced your amplicons on an Illumina instrument, the sequencing platform is essentially irrelevant).*

We have clarified our strategic approach and changed "Illumina" to "shotgun" as correctly highlighted by the reviewer.

- *Line 145: Provide basic statistics comparing the 19 full-length genomes before delving into more elaborate phylogenomic analyses (as suggested above).*

These were already provided and discussed in now supplementary figure S10.

- *Lines 163-165: The last sentence can be removed as readers are typically familiar with scales.*

This sentence has been removed

- *Figure 2: The labels in the trees are difficult to read and could be replaced with simple colored squares or symbols. Enlarging the trees or providing zoomed-in sections, particularly of the areas representing FCoV-23, would enhance readability.*

This figure has now been updated.

- *Lines 196-199: This analysis may not add substantial value as it reports averages of separate evolutionary trajectories. Consider focusing on more informative phylodynamic analyses as suggested earlier.*

We have retained this analysis as well as added further phylodynamic analyses as suggested.

- *Lines 272-273: If no other cases abroad have been reported thus far, mention this to suggest that the actual risk may be low.*

We have updated this section with the most recent information highlighting further cases.

- *Line 304: Remove "earlier discussed".*

Removed

- *Lines 308-319: This paragraph repeats earlier arguments. Consider integrating the few new points into previous sections to avoid redundancy, and delete it here entirely.*

We believe this is a worthwhile summary to end the paper and have retained this section.